# Boosting Dilated Convolutional Networks with Mixed Tensor Decompositions

**Nadav Cohen**
Institute for Advanced Study
`cohennadav@ias.edu`

**Ronen Tamari**
The Hebrew University of Jerusalem
`ronent@cs.huji.ac.il`

**Amnon Shashua**
The Hebrew University of Jerusalem
`shashua@cs.huji.ac.il`

## ABSTRACT

The driving force behind deep networks is their ability to compactly represent rich classes of functions. The primary notion for formally reasoning about this phenomenon is expressive efficiency, which refers to a situation where one network must grow unfeasibly large in order to replicate functions of another. To date, expressive efficiency analyses focused on the architectural feature of depth, showing that deep networks are representationally superior to shallow ones. In this paper we study the expressive efficiency brought forth by connectivity, motivated by the observation that modern networks interconnect their layers in elaborate ways. We focus on dilated convolutional networks, a family of deep models delivering state of the art performance in sequence processing tasks. By introducing and analyzing the concept of mixed tensor decompositions, we prove that interconnecting dilated convolutional networks can lead to expressive efficiency. In particular, we show that even a single connection between intermediate layers can already lead to an almost quadratic gap, which in large-scale settings typically makes the difference between a model that is practical and one that is not. Empirical evaluation demonstrates how the expressive efficiency of connectivity, similarly to that of depth, translates into gains in accuracy. This leads us to believe that expressive efficiency may serve a key role in developing new tools for deep network design.

## 1 INTRODUCTION

One of the key attributes fueling the success of deep learning is the ability of deep networks to compactly represent rich classes of functions. This phenomenon has drawn considerable attention from the theoretical machine learning community in recent years. The primary notion for formally reasoning about the representational abilities of different models is *expressive efficiency*. Given two network architectures $A$ and $B$, with size parameters (typically the width of layers across a network) $r_A$ and $r_B$, we say that architecture $A$ is expressively efficient w.r.t. architecture $B$ if the following two conditions hold: *(i)* any function realized by $B$ with size $r_B$ can be realized (or approximated) by $A$ with size $r_A \in \mathcal{O}(r_B)$; *(ii)* there exist functions realized by $A$ with size $r_A$ that cannot be realized (or approximated) by $B$ unless its size meets $r_B \in \Omega(f(r_A))$ for some super-linear function $f$. The nature of the function $f$ in condition *(ii)* determines the type of efficiency taking place – if $f$ is exponential then architecture $A$ is said to be exponentially expressively efficient w.r.t. architecture $B$, and if $f$ is polynomial so is the expressive efficiency of $A$ over $B$.

To date, works studying expressive efficiency in the context of deep learning (e.g. Delalleau and Bengio (2011); Pascanu et al. (2013); Montufar et al. (2014); Telgarsky (2015); Eldan and Shamir (2015); Poole et al. (2016); Raghu et al. (2016); Cohen et al. (2016b); Cohen and Shashua (2016); Poggio et al. (2015); Mhaskar et al. (2016)) have focused on the architectural feature of depth, showing instances where deep networks are expressively efficient w.r.t. shallow ones. This theoretical focus is motivated by the vast empirical evidence supporting the importance of depth (*cf.* LeCun et al. (2015)). However, it largely overlooks an additional architectural feature that in recent years is proving to have great impact on the performance of deep networks – *connectivity*. Nearly all state of the art networks these days (e.g. Szegedy et al. (2015); He et al. (2015); Huang et al. (2016b;a))

deviate from the simple feed-forward (chain) approach, running layers connected under various schemes. Whether or not this relates to expressive efficiency remains to be an open question.

A specific family of deep networks gaining increased attention in the deep learning community is that of *dilated convolutional networks*. These models form the basis of the recent WaveNet (van den Oord et al. (2016)) and ByteNet (Kalchbrenner et al. (2016)) architectures, which provide state of the art performance in audio and text processing tasks. Dilated convolutional networks are frequently applied to sequence data, and consist of multiple succeeding convolutional layers, each comprising non-contiguous filters with a different dilation (distance between neighboring elements). The choice of dilations directly affects the space of functions that may be realized by a network, and while no choice is expressively efficient w.r.t. another, we show in this work that interconnecting networks with different dilations leads to expressive efficiency, and by this demonstrate that connectivity indeed bears the potential to enhance the expressiveness of deep networks.

Our analysis follows several recent works utilizing tensor decompositions for theoretical studies of deep learning (*e.g.* Janzamin et al. (2015); Sedghi and Anandkumar (2016)), and in particular, builds on the equivalence between hierarchical tensor decompositions and convolutional networks established in Cohen et al. (2016b) and Cohen and Shashua (2016). We show that with dilated convolutional networks, the choice of dilations throughout a network corresponds to determination of the mode (dimension) tree underlying the respective decomposition. We then define the notion of a *mixed tensor decomposition*, which blends together multiple mode trees, effectively creating a large ensemble of hybrid trees formed from all possible combinations. Mixed tensor decompositions correspond to mixed dilated convolutional networks, *i.e.* mixtures formed by connecting intermediate layers of different dilated convolutional networks. This allows studying the expressive properties of such mixtures using mathematical machinery from the field of tensor analysis. We fully analyze a particular case of dilated convolutional arithmetic circuits, showing that a single connection between intermediate layers already leads to an almost quadratic expressive efficiency, which in large-scale settings typically makes the difference between a model that is practical and one that is not.

An experiment on TIMIT speech corpus (Garofolo et al. (1993)) evaluates the dilated convolutional network architectures covered by our analysis. We find that interconnecting intermediate layers of different networks improves accuracy, with no additional cost in terms of computation or model capacity. This serves as an indication that with the architectural feature of connectivity, similarly to the case of depth, expressive efficiency and improved accuracies go hand in hand. Accordingly, we believe expressive efficiency may serve a key role in developing new tools for deep network design.

## 2 SUMMARY OF OUR ANALYSIS AND CONTRIBUTIONS

For the convenience of the reader, we summarize below the analysis and contributions of this paper. The summarized material is delivered fully in sec. 3, 4, 5 and the appendices referenced therein. To keep the manuscript at reasonable length, much of the material is located in the appendices. We refer the reader to Cohen et al. (2017) for a longer, self-contained version of the text.

Our analysis begins in sec. 3, where we present the dilated convolutional network underlying WaveNet (fig. 1). We consider this to be the baseline architecture and, following Cohen and Shashua (2016), facilitate its study through tensor analysis. The key to introducing tensors into the framework is a discretization of the network's input-output mapping. Namely, $f(\mathbf{x}[t-N+1], \ldots, \mathbf{x}[t]) - a$ function realized by the network ($t$ here stands for a natural time index), is conceptually evaluated on a finite (exponentially large) number of input points, generated from all possible assignments of the variables $\mathbf{x}[t-N+1], \ldots, \mathbf{x}[t]$ to each hold one of $M$ predetermined values. This gives rise to an $N$-dimensional lookup table, with length $M$ in each axis. We refer to this lookup table as a *grid tensor* (eq. 1). It is shown (app. C) that grid tensors brought forth by the baseline dilated convolutional network (fig. 1) can be expressed as a hierarchical tensor decomposition, referred to as the *baseline decomposition* (eq. 2).

The baseline decomposition implicitly adheres to a particular tree over tensor modes (axes). This calls for a generalization, and we indeed define a general mode tree (def. 1), followed by a corresponding hierarchical tensor decomposition, referred to as the *tree decomposition* (eq. 3). Different choices of mode trees lead to tree decompositions characterizing networks with different dilations. We focus on the tree that corresponds to the baseline network (fig. 2(a)), and on those corresponding to networks obtained by swapping dilations of different layers (fig. 2(b), for example).

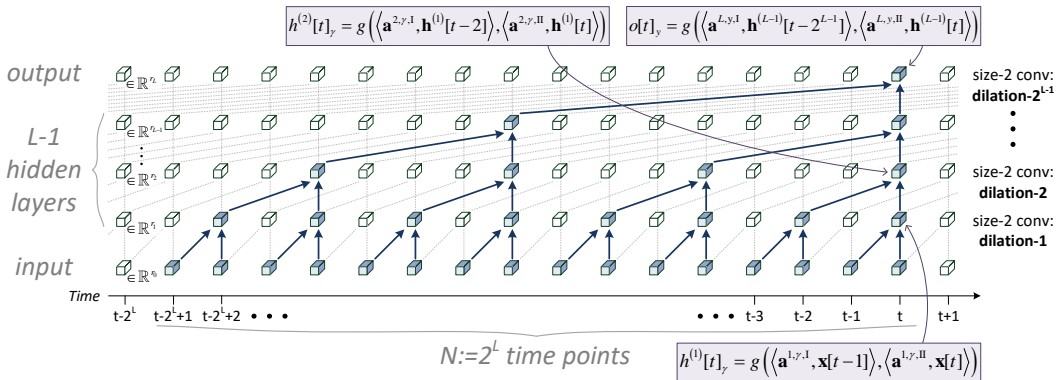

Figure 1: Baseline dilated convolutional network architecture (see description in app. B).

Armed with a framework for representing different dilated convolutional networks through hierarchical tensor decompositions of different mode trees, we head on in sec. 4 and introduce the notion of a *mixed tensor decomposition* (eq. 4). The mixed decomposition of two mode trees $T$ and $\bar{T}$ is based on a preselected set of nodes present in both trees, referred to as *mixture nodes*. Individual tree decompositions of $T$ and $\bar{T}$ are run in parallel, where at each mixture node, tensors from the two decompositions are swapped. If $\mathcal{N}$ and $\bar{\mathcal{N}}$ are the dilated convolutional networks characterized by $T$ and $\bar{T}$ (respectively), the mixed decomposition characterizes a mixed (interconnected) network $\mathcal{M}$, formed by rewiring intermediate layers of $\mathcal{N}$ into $\bar{\mathcal{N}}$, and vice versa (see illustration in fig. 3).

The heart of our analysis is sec. 5, where we study the expressive efficiency of the mixed network $\mathcal{M}$ over the individual networks $\mathcal{N}$ and $\bar{\mathcal{N}}$. Establishing expressive efficiency requires showing that any function realized by $\mathcal{N}$ or $\bar{\mathcal{N}}$ can be realized by $\mathcal{M}$ with no more than linear growth in size, whereas the converse does not hold, *i.e.* there exist functions realizable by $\mathcal{M}$ that cannot be realized by $\mathcal{N}$ or $\bar{\mathcal{N}}$ unless their size is allowed to grow super-linearly. From a tensor decomposition perspective, this translates to the following two propositions:

- *(i)* any tensor generated by a tree decomposition of $T$ or $\bar{T}$ can be realized by their mixed decomposition with no more than linear growth in size;

- *(ii)* there exist tensors realizable by the mixed decomposition of $T$ and $\bar{T}$ that cannot be realized by their individual tree decompositions without a super-linear growth in size.

We address both propositions through the notion of *hybrid mode trees* (def. 2; fig. 4), which are simply mode trees born from combinations of $T$ and $\bar{T}$. We prove (claim 1) that the mixed decomposition of $T$ and $\bar{T}$ can replicate, with no more than linear growth in size, the tree decomposition of any hybrid tree $H$. Since $T$ and $\bar{T}$ are in particular hybrid mode trees of themselves, we obtain an affirmative answer to proposition *(i)*. For addressing proposition *(ii)*, we demonstrate a case (with convolutional arithmetic circuits) where there exists a hybrid tree $H$ whose tree decomposition generates tensors that require the tree decompositions of $T$ and $\bar{T}$ to grow super-linearly. Since the mixed decomposition of $T$ and $\bar{T}$ can (by claim 1) replicate the tree decomposition of $H$ with no more than linear growth, proposition *(ii)* is established, and $\mathcal{M}$ is indeed expressively efficient w.r.t. $\mathcal{N}$ and $\bar{\mathcal{N}}$ (corollary 1).

The central tool for establishing proposition *(ii)*, or more specifically, for demonstrating the existence of a hybrid tree $H$ whose tree decomposition requires those of $T$ and $\bar{T}$ to grow super-linearly, is a tight analysis of tensors generated by a tree decomposition in terms of their ranks when arranged as matrices (theorem 1). Matricization ranks under hierarchical tensor decompositions are of interest from a pure tensor analysis perspective (*cf.* Hackbusch (2012)), as well as in the context of deep learning (*cf.* Cohen and Shashua (2017)). The bounds we provide are much tighter (exact in many cases) and far more general than those existing in the literature, and we expect them to prove useful in different applications. The key idea in deriving these bounds is to consider a matricized form of the tree decomposition, and recursively propagate outwards various matrices (for details see proof of theorem 1 in app. E.2).

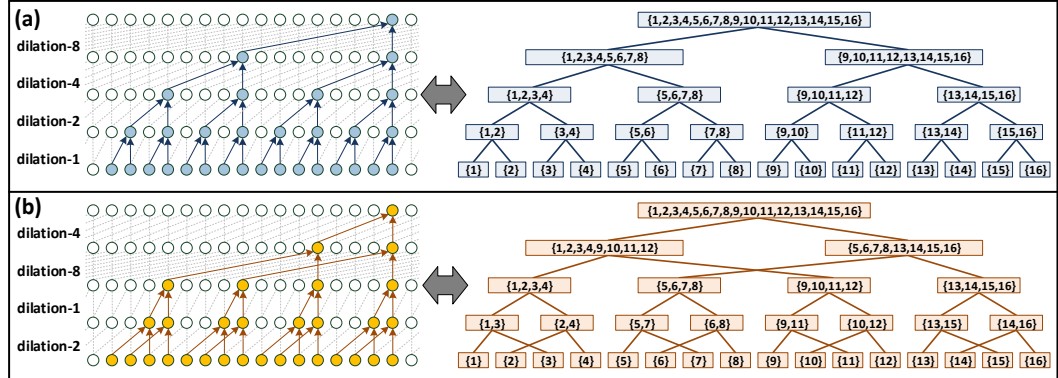

Figure 2: Best viewed in color. Dilated convolutional networks (left) and the mode trees underlying their respective tensor decompositions (right). *(a)* Baseline architecture – dilation $2^{l-1}$ in layer $l$. *(b)* Architecture obtained by swapping dilations of even and odd layers.

To conclude this section, we list below the main contributions of the paper:

- We introduce the notion of a mixed tensor decomposition, and prove that it brings forth a representational advantage compared to the individual hierarchical decompositions it comprises. This development is of interest from a pure tensor analysis perspective, independently of convolutional networks, or machine learning in general.

- We provide the first formal evidence for the fact that interconnectivity – an architectural feature prevalent in state of the art deep learning, brings forth expressive efficiency.

- Our central theorem (theorem 1) provides the most comprehensive characterization to date of matricization ranks brought forth by hierarchical tensor decompositions.

## 3 DILATED CONVOLUTIONAL NETWORKS

*Dilated convolutional networks* are a family of convolutional networks (LeCun and Bengio (1995)) gaining increased attention in the deep learning community. As opposed to more conventional convolutional architectures (*e.g.* Krizhevsky et al. (2012)), which are applied primarily to images (and videos), dilated convolutional networks thrive in sequence processing tasks. For example, they underlie Google's WaveNet (van den Oord et al. (2016)) and ByteNet (Kalchbrenner et al. (2016)) models, which provide state of the art performance in audio and text processing tasks.

### 3.1 BASELINE ARCHITECTURE

The dilated convolutional network architecture considered as baseline in this paper is the one underlying WaveNet, depicted in fig. 1. Due to lack of space, we defer its detailed description to app. B, and merely note here that we use $g(\cdot)$ to denote the function combining two size-1 convolutions into a single size-2 convolution with non-linearity (*e.g.* $g(a,b):=\max\{a+b,0\}$ for ReLU activation).

Our interest lies on the representational abilities of a network, *i.e.* on the properties of the input-output mappings it can realize. For a fixed time point $t$, $\mathbf{o}[t]$ – network output at time $t$, is a function of $\mathbf{x}[t-2^L+1]\ldots\mathbf{x}[t]$ – network input over the last $2^L$ time points. Taking into account temporal stationarity, and denoting for brevity $N:=2^L$, we may write $o[t]_y = f_y(\mathbf{x}[t-N+1],\ldots,\mathbf{x}[t])$ for every output coordinate $y \in [r_L]$. We study the functions $\{f_y(\cdot)\}_y$, which obviously depend on the convolution weights $\{\mathbf{a}^{l,\gamma,\mathrm{I}},\mathbf{a}^{l,\gamma,\mathrm{II}}\}_{l,\gamma}$, through the process of *discretization*. Namely, we choose a collection of vectors $\mathbf{v}^{(1)}\ldots\mathbf{v}^{(M)}$, and for each output coordinate $y$, define the following tensor:

$$\mathcal{A}^y_{d_1\ldots d_N} := f_y(\mathbf{v}^{(d_1)},\ldots,\mathbf{v}^{(d_N)}) \quad \forall d_1\ldots d_N \in [M] \tag{1}$$

$\mathbf{v}^{(1)}\ldots\mathbf{v}^{(M)}$ are referred to as *discretizers*, and $\mathcal{A}^y$ is referred to as the *grid tensor* of $f_y(\cdot)$. The size of a grid tensor is exponential in $N$, thus treating it directly is intractable. However, the network admits a compact parameterization of grid tensors in terms of its convolution weights (see app. C, and the preliminaries in app. A):

For $j = 1 \ldots N$:
$$\phi^{0,j,\gamma} = [v_\gamma^{(1)}, \ldots, v_\gamma^{(M)}]^\top \quad \forall \gamma \in [r_0]$$
For $l = 1 \ldots L$, $j = 1 \ldots N/2^l$:
$$\phi^{l,j,\gamma} = \left( \sum_{\alpha=1}^{r_{l-1}} a_\alpha^{l,\gamma,\mathrm{I}} \cdot \phi^{l-1,2j-1,\alpha} \right) \otimes_g \left( \sum_{\alpha=1}^{r_{l-1}} a_\alpha^{l,\gamma,\mathrm{II}} \cdot \phi^{l-1,2j,\alpha} \right) \quad \forall \gamma \in [r_l]$$
$$\mathcal{A}^y = \phi^{L,1,y} \quad \forall y \in [r_L] \tag{2}$$

This parameterization is in fact a hierarchical tensor decomposition. To highlight its correspondence to the baseline dilated convolutional network (fig. 1), we refer to it as the *baseline decomposition*.

## 3.2 Dilations and Mode Trees

The baseline decomposition (eq. 2), corresponding to the baseline dilated convolutional network (fig. 1), implicitly adheres to a tree structure[1] In this subsection we generalize the underlying tree, and show that the resulting decompositions capture networks with various dilations throughout their convolutional layers. We begin by defining a general (binary) tree over tensor modes:

**Definition 1.** *Let $N \in \mathbb{N}$. A* binary mode tree[2] *over $[N]$ is a full binary tree[3] in which:*

- *Every node is labeled by a subset of $[N]$*
- *There are exactly $N$ leaves, labeled $\{1\} \ldots \{N\}$*
- *The label of an interior (non-leaf) node is the union of the labels of its children*

*If $T$ is a binary mode tree, we identify its nodes with their labels,* i.e. *with the corresponding subsets of $[N]$. The set of all interior nodes is denoted by $int(T) \subset 2^{[N]}$; the children of an interior node $\nu \subset [N]$ are denoted by $C_I(\nu; T), C_{II}(\nu; T) \subset [N]$; and the parent of a non-root node $\nu \subset [N]$ is denoted by $P(\nu; T)$. Notice that by definition, the root node is labeled $[N]$.*

Recall the definition of grid tensors $\{\mathcal{A}^y\}_y$ (eq. 1), and let $T$ be a binary mode tree over $[N]$. $T$ induces a hierarchical decomposition of the grid tensors, referred to as its *tree decomposition*:

For $j = 1 \ldots N$:
$$\phi^{\{j\},\gamma} = [v_\gamma^{(1)}, \ldots, v_\gamma^{(M)}]^\top \quad \forall \gamma \in [r]$$
For $\nu$ in $int(T)$ (depth-first order):
$$\phi^{\nu,\gamma} = \sigma^{(\nu;T)} \left( \left( \sum_{\alpha=1}^{r} a_\alpha^{\nu,\gamma,\mathrm{I}} \cdot \phi^{C_I(\nu;T),\alpha} \right) \otimes_g \left( \sum_{\alpha=1}^{r} a_\alpha^{\nu,\gamma,\mathrm{II}} \cdot \phi^{C_{II}(\nu;T),\alpha} \right) \right) \quad \forall \gamma \in [r]$$
$$\mathcal{A}^y = \phi^{[N],y} \quad \forall y \in [r] \tag{3}$$

To conserve space we defer the annotation of the tree decomposition to app. D, noting that $r \in \mathbb{N}$ – the number of tensors in each group $\{\phi^{\nu,\gamma}\}_\gamma$, is referred to as the *size constant* of the decomposition.[4]

Compare the general tree decomposition (eq. 3) to the baseline decomposition (eq. 2). It is not difficult to see that the latter is a special case of the former, corresponding to a binary mode tree $T$ that is perfect,[5] and whose depth-$l$ nodes are adjacent sets of size $N/2^l$ (fig. 2(a)-right). This implies that such a mode tree, when plugged into the tree decomposition, provides a characterization of the baseline dilated convolutional network (fig. 1), *i.e.* a network whose dilation in layer $l$ is $2^{l-1}$ (fig. 2(a)-left). If we were to choose a different mode tree, the corresponding dilated convolutional network would change.[6] For example, if we swap connections in the mode tree (fig. 2(b)-right), we obtain a decomposition that characterizes a network whose dilations are swapped (fig. 2(b)-left).

## 4 Mixed Tensor Decompositions

Let $T$ and $\bar{T}$ be two binary mode trees over $[N]$ (def. 1). We will now define *mixed tensor decompositions*, blending together the tree decompositions of $T$ and $\bar{T}$ (eq. 3). A mixed decomposition of $T$ and $\bar{T}$ is obtained by choosing a collection of *mixture nodes* $mix(T, \bar{T})$. These are nodes (subsets of $[N]$) that reside in the interior of both $T$ and $\bar{T}$, defining locations in the tree decompositions at which tensors will be exchanged. If $mix(T, \bar{T})$ is chosen as the empty set, the mixed decomposition simply sums the output tensors generated by the tree decompositions of $T$ and $\bar{T}$. Otherwise, the tree decompositions of $T$ and $\bar{T}$ progress in parallel, until reaching a mixture node $\mu \in mix(T, \bar{T})$, where they exchange tensors between them. The process continues until all mixture nodes are visited and the root node (of both trees) $[N]$ is reached. At this point tensors are summed and returned as output. The formal definition of the mixed decomposition, annotated in detail in app. D, is as follows:

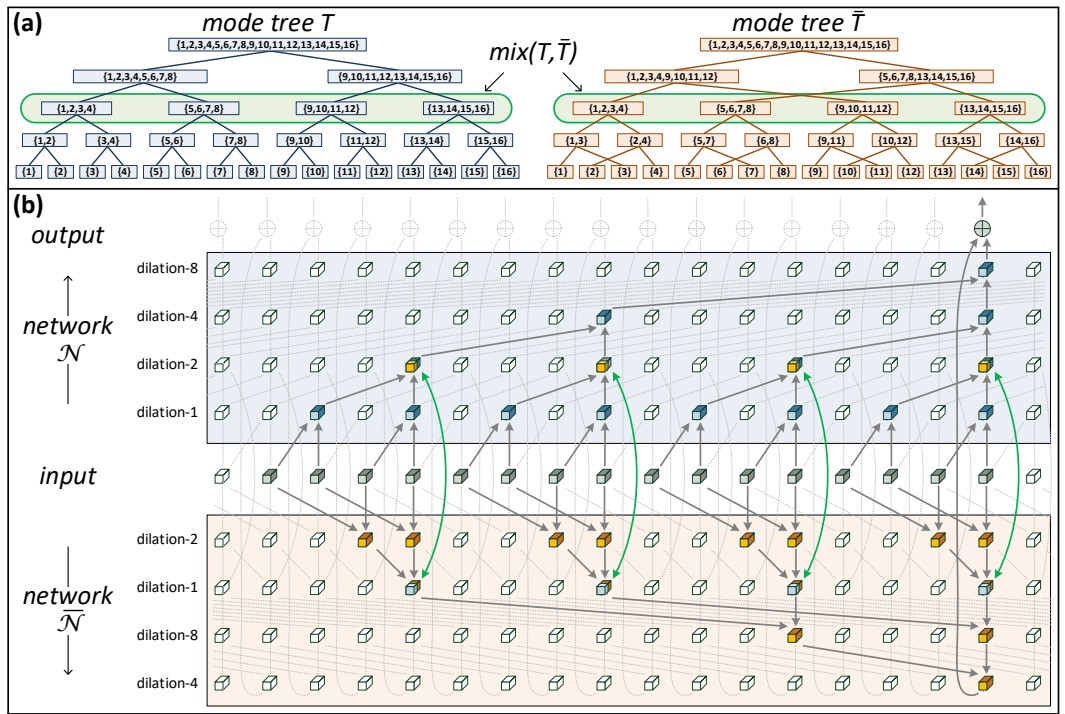

Figure 3: To be viewed in color. **(a)** Two mode trees $T$ and $\bar{T}$ (given on the right of fig. 2), along with a possible choice of mixture nodes $mix(T, \bar{T})$ for the mixed decomposition (eq. 4). **(b)** Mixed dilated convolutional network corresponding to chosen mixed decomposition. Networks $\mathcal{N}$ and $\bar{\mathcal{N}}$ associated with $T$ and $\bar{T}$ (fig. 2, left) are combined through output summation and rewiring of an intermediate convolutional layer (green).

$1:$ For $j = 1 \ldots N$:

$2:$ $\quad \phi^{\{j\},\gamma} = \bar{\phi}^{\{j\},\gamma} = [v_\gamma^{(1)}, \ldots, v_\gamma^{(M)}]^\top \quad \forall \gamma \in [r]$ $\qquad$ $\boxed{r - \text{decomposition size constant}}$

$3:$ For $\mu$ in $mix(T, \bar{T}) \cup \{[N]\}$ (inclusion order):

$4:$ $\quad$ For $\nu$ in $int(T) \cap 2^\mu \setminus \{\text{nodes in } T \text{ already visited}\}$ (inclusion order):

$5:$ $\quad\quad \phi^{\nu,\gamma} = \sigma^{(\nu;T)} \left( \left( \sum_{\alpha=1}^r a_\alpha^{\nu,\gamma,\mathrm{I}} \cdot \phi^{C_\mathrm{I}(\nu;T),\alpha} \right) \otimes_g \left( \sum_{\alpha=1}^r a_\alpha^{\nu,\gamma,\mathrm{II}} \cdot \phi^{C_\mathrm{II}(\nu;T),\alpha} \right) \right) \quad \forall \gamma \in [r]$

$6:$ $\quad$ For $\bar{\nu}$ in $int(\bar{T}) \cap 2^\mu \setminus \{\text{nodes in } \bar{T} \text{ already visited}\}$ (inclusion order):

$7:$ $\quad\quad \bar{\phi}^{\bar{\nu},\gamma} = \sigma^{(\bar{\nu};\bar{T})} \left( \left( \sum_{\alpha=1}^r \bar{a}_\alpha^{\bar{\nu},\gamma,\mathrm{I}} \cdot \bar{\phi}^{C_\mathrm{I}(\bar{\nu};\bar{T}),\alpha} \right) \otimes_g \left( \sum_{\alpha=1}^r \bar{a}_\alpha^{\bar{\nu},\gamma,\mathrm{II}} \cdot \bar{\phi}^{C_\mathrm{II}(\bar{\nu};\bar{T}),\alpha} \right) \right) \quad \forall \gamma \in [r]$

$8:$ $\quad$ Swap $\phi^{\mu,\gamma} \longleftrightarrow \bar{\phi}^{\mu,\gamma} \quad \forall \gamma \in [r/2]$

$9:$ $\mathcal{A}^y = \phi^{[N],y} + \bar{\phi}^{[N],y} \quad \forall y \in [r]$ $\hfill$ (4)

Let $\mathcal{N}$ and $\bar{\mathcal{N}}$ be the dilated convolutional networks whose input-output mappings are characterized by the tree decompositions of $T$ and $\bar{T}$ (respectively). The mixed decomposition of $T$ and $\bar{T}$ (eq. 4) characterizes the input-output mapping of a *mixed dilated convolutional network*, formed by summing the outputs of $\mathcal{N}$ and $\bar{\mathcal{N}}$, and interconnecting their intermediate layers. The choice of mixture nodes $mix(T, \bar{T})$ effectively determines the locations at which networks $\mathcal{N}$ and $\bar{\mathcal{N}}$ are interconnected, where an interconnection simply wires into $\mathcal{N}$ outputs of a convolutional layer in $\bar{\mathcal{N}}$, and vice versa. For example, suppose that $\mathcal{N}, \bar{\mathcal{N}}, T$ and $\bar{T}$ are the networks and trees portrayed in fig. 2. A possible choice of mixture nodes, and the resulting mixed network, are illustrated in fig. 3.

## 5 EXPRESSIVE EFFICIENCY ANALYSIS

As in sec. 4, let $\mathcal{N}$ and $\bar{\mathcal{N}}$ be two dilated convolutional networks whose input-output mappings are characterized by the tree decomposition (eq. 3) with mode trees $T$ and $\bar{T}$ respectively. Consider the mixed decomposition (eq. 4) resulting from a particular choice of mixture nodes $mix(T, \bar{T})$, and denote its corresponding mixed dilated convolutional network by $\mathcal{M}$. We would like to show that $\mathcal{M}$ is expressively efficient w.r.t. $\mathcal{N}$ and $\bar{\mathcal{N}}$. This amounts to addressing the following two propositions:[7]

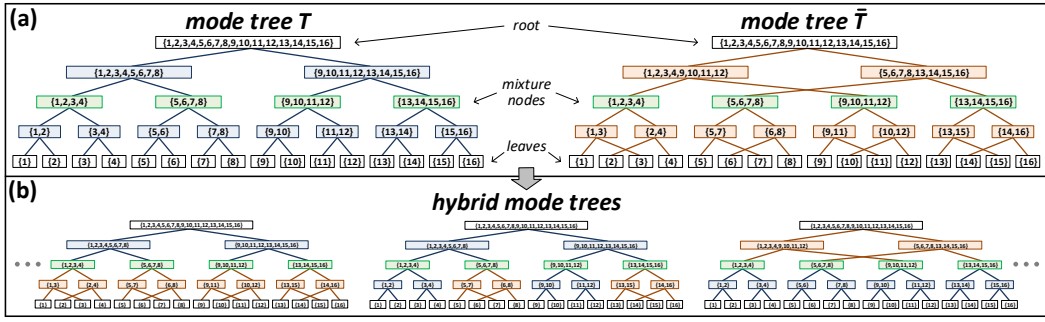

Figure 4: Best viewed in color. *(a)* Two mode trees $T$ and $\bar{T}$ along with a possible choice of mixture nodes (same as in fig. 3(a)). *(b)* Sample of the resulting hybrid mode trees (def. 2).

**Proposition 1.** *Consider a tree decomposition (eq. 3) with underlying mode tree $T$ or $\bar{T}$ and size constant $r = r_{tree}$. This decomposition can be realized by a mixed decomposition of $T$ and $\bar{T}$ (eq. 4) whose size constant $r$ is linear in $r_{tree}$.*

**Proposition 2.** *Consider a mixed decomposition of $T$ and $\bar{T}$ (eq. 4) with size constant $r = r_{mix}$. This decomposition can generate grid tensors $\{\mathcal{A}^y\}_y$ that cannot be generated by tree decompositions of $T$ or $\bar{T}$ (eq. 3) unless their size constant $r$ is super-linear in $r_{mix}$.*

As a first step in treating prop. 1 and 2, we define the notion of a hybrid mode tree:

**Definition 2.** *Let $T$ and $\bar{T}$ be binary mode trees over $[N]$ (def. 1), and let $mix(T, \bar{T})$ be a corresponding collection of mixture nodes, i.e. a set of nodes (subsets of $[N]$) contained in the interior of both $T$ and $\bar{T}$. We say that $H$ is a hybrid mode tree of $T$ and $\bar{T}$ w.r.t. $mix(T, \bar{T})$, if it is a binary mode tree over $[N]$, whose interior may be generated by the following process:*

$$int(H) = \emptyset$$
$$\text{For } \mu \text{ in } mix(T, \bar{T}) \cup \{[N]\} \text{ (inclusion order):}$$
$$S = int(T) \cap 2^\mu \setminus \{\text{nodes in } T \text{ already assigned to } S\}$$
$$\bar{S} = int(\bar{T}) \cap 2^\mu \setminus \{\text{nodes in } \bar{T} \text{ already assigned to } \bar{S}\}$$
$$int(H) = int(H) \cup S \quad \textbf{or} \quad int(H) = int(H) \cup \bar{S}$$

*In words, for every $\mu$ that is either a mixture node or the root node, $int(H)$ includes a* segment *from either $int(T)$ or $int(\bar{T})$, where the segment comprises all descendants of $\mu$ from which the path to $\mu$ does not cross any other mixture node (see illustration in fig. 4).*

Claim 1 below states that with proper weight setting, a mixed decomposition of $T$ and $\bar{T}$ (eq. 4) with size constant $r=r_{mix}$, can realize any tree decomposition (eq. 3) with size constant $r=r_{mix}/2$, if the underlying mode tree is a hybrid of $T$ and $\bar{T}$. Since $T$ and $\bar{T}$ are in particular hybrid mode trees of themselves, we obtain an affirmative answer to prop. 1.

**Claim 1** (proof in app. E.1)**.** *Let $T$ and $\bar{T}$ be binary mode trees over $[N]$ (def. 1), and let $mix(T, \bar{T})$ be a corresponding collection of mixture nodes. Consider a mixed decomposition of $T$ and $\bar{T}$ w.r.t. $mix(T, \bar{T})$ (eq. 4), and denote its size constant $r$ by $r_{mix}$. Let $H$ be a hybrid mode tree of $T$ and $\bar{T}$ w.r.t. $mix(T, \bar{T})$ (def. 2), and consider the respective tree decomposition (eq. 3), with size constant $r=r_{mix}/2$. For any setting of weights $\{\mathbf{a}^{\nu,\gamma,I}, \mathbf{a}^{\nu,\gamma,II}\}_{\nu,\gamma}$ leading to grid tensors $\{\mathcal{A}^y\}_y$ in this tree decomposition, there exists a setting of weights $\{\mathbf{a}^{\nu,\gamma,I}, \mathbf{a}^{\nu,\gamma,II}\}_{\nu,\gamma}, \{\bar{\mathbf{a}}^{\bar{\nu},\gamma,I}, \bar{\mathbf{a}}^{\bar{\nu},\gamma,II}\}_{\bar{\nu},\gamma}$ in the mixed decomposition, independent of discretizers $\{\mathbf{v}^{(i)}\}_{i\in[M]}$, that leads to the same grid tensors.*[8]

Claim 1 not only addresses prop. 1, but also brings forth a strategy for treating prop. 2. The strategy is to find a hybrid mode tree $H$ distinct enough from $T$ and $\bar{T}$ such that its tree decomposition, which according to claim 1 is easily realized by the mixed decomposition, poses a significant challenge for the individual tree decompositions of $T$ and $\bar{T}$. Hereinafter we pursue this line of reasoning, focusing on the particular case of convolutional arithmetic circuits – $g(a,b)=a{\cdot}b$. We focus on this special case since it allows the use of a plurality of algebraic tools for theoretical analysis, while at the same time corresponding to models showing promising results in practice (see for example Cohen et al. (2016a); Sharir et al. (2016)).[9]

To crisply phrase our central theorem, we define the notion of an index set tiled by a mode tree:

**Definition 3.** *Let $T$ be a binary mode tree over $[N]$ (def. 1), and let $\mathcal{I} \subset [N]$ be a non-empty set of indexes. A tiling of $\mathcal{I}$ by $T$ is a collection of nodes in the tree, denoted $\Theta(\mathcal{I}; T)$, which meets*

*the following two requirements:* (i) $\cup_{\nu \in \Theta(\mathcal{I};T)} \nu = \mathcal{I}$ (ii) $\nu \in \Theta(\mathcal{I};T) \Rightarrow P(\nu;T) \not\subset \mathcal{I}$. *In words,* $\Theta(\mathcal{I};T)$ *is a set of nodes in* $T$ *whose disjoint union gives* $\mathcal{I}$, *where each node is maximal,* i.e. *its parent in the tree is not a subset of* $\mathcal{I}$. *See illustration in fig. 6 (supplementary material).*

Theorem 1 below provides a tight characterization of grid tensors generated by a tree decomposition in terms of their ranks when matricized (see app. A) w.r.t. an index set. This result is of general interest from both tensor analysis and deep learning perspectives. We use it to establish prop. 2.

**Theorem 1** (proof in app. E.2). *Let* $T$ *be a binary mode tree over* $[N]$ *(def. 1), and consider the corresponding tree decomposition (eq. 3) with discretizers* $\mathbf{v}^{(1)} \ldots \mathbf{v}^{(M)}$ *spanning* $\mathbb{R}^r$. *Assume that* $g(a,b) = a \cdot b$ *(non-generalized decomposition – see app. A), and suppose the generated grid tensors* $\{\mathcal{A}^y\}_y$ *are matricized (see app. A) w.r.t. an index set* $\mathcal{I} \subset [N]$, $\emptyset \neq \mathcal{I} \neq [N]$, *whose complement we denote by* $\mathcal{I}^c := [N] \setminus \mathcal{I}$. *Then, the ranks of the grid tensor matricizations* $\{ [\![\mathcal{A}^y]\!]_{\mathcal{I}} \}_y$ *are:*

- *no greater than* $r^{\min\{|\Theta(\mathcal{I};T)|, |\Theta(\mathcal{I}^c;T)|\}}$
- *at least* $r^{|\{(\nu_1,\nu_2) \in \Theta(\mathcal{I};T) \times \Theta(\mathcal{I}^c;T): \ \nu_1 \text{ and } \nu_2 \text{ are siblings in } T \text{ with depth}>1\}|}$ *almost always,* i.e. *for all configurations of weights* $\{\mathbf{a}^{\nu,\gamma,I}, \mathbf{a}^{\nu,\gamma,II}\}_{\nu,\gamma}$ *but a set of Lebesgue measure zero*

Given two mode trees $T$ and $\bar{T}$, with a corresponding collection of mixture nodes $mix(T, \bar{T})$, the bounds in theorem 1 can be used to find an index set $\mathcal{I}$ and a hybrid mode tree $H$, such that the tree decomposition of $H$ generates grid tensors whose ranks under matricization w.r.t. $\mathcal{I}$ are much higher than those brought forth by the tree decompositions of $T$ and $\bar{T}$. This fulfills the strategy described above, thereby establishing prop. 2. In app. F we demonstrate this process with the exemplar setting considered throughout the paper (fig. 2, 3, 4). The following corollary is reached:

**Corollary 1.** *Let* $\mathcal{N}$ *be the baseline dilated convolutional network (fig. 1), and let* $\bar{\mathcal{N}}$ *be a network obtained by swapping dilations of groups of* $k$ *layers (the case* $k=2$ *is illustrated in fig. 2(b)-left). Denote by* $\mathcal{M}$ *the mixed network obtained by summing the outputs of* $\mathcal{N}$ *and* $\bar{\mathcal{N}}$, *while interconnecting their* $k$'*th intermediate layer (and possibly additional layers). Assume the networks' convolutional operator* $g(\cdot)$ *is a product. Then, besides a negligible set, all functions realized by* $\mathcal{M}$ *with* $r$ *channels in the layers of each interconnected network, cannot be realized by* $\mathcal{N}$ *(or* $\bar{\mathcal{N}}$*) if the number of channels in each of its layers is less than* $(r/2)^{2/(1+2^{1-k})}$.

Corollary 1 (along with claim 1) demonstrates that interconnecting intermediate layers of different dilated convolutional networks can bring forth expressive efficiency. The lower bound in the corollary – $(r/2)^{2/(1+2^{1-k})}$, is essentially quadratic when $k \geq 4$. For example, if $k = 4$ and the number of channels $r$ in each interconnected network is 128, the lower bound implies that in order to maintain representational abilities with an individual network, over 1500 channels in each layer are required – far beyond acceptable practice in deep learning.

## 6 EXPERIMENT

To assess the practical implications of the expressive efficiency brought forth by mixing dilated convolutional networks, a simple experiment was conducted. We trained a baseline dilated convolutional network $\mathcal{N}$ (dilation $2^{l-1}$ in layer $l$ – see sec. 3.1) with architectural parameters similar to those used in WaveNet (van den Oord et al. (2016)), to classify individual phonemes in the TIMIT acoustic speech corpus (Garofolo et al. (1993)). In addition to the baseline model, we also trained a companion network $\bar{\mathcal{N}}$ obtained by swapping dilations of even and odd layers. The mode trees corresponding to these networks (illustrated in fig. 2) – $T$ and $\bar{T}$, share interior nodes of even depth, thus any subset of those nodes may serve as mixture nodes for a mixed decomposition (eq. 4). We evaluate mixed dilated convolutional networks $\mathcal{M}$ corresponding to different choices of mixture nodes (see fig. 3 for illustration of a particular case). Specifically, we consider choices of the following form: $mix(T, \bar{T}) := \{\nu \in int(T) \cap int(\bar{T}) : \text{depth of } \nu \text{ (in } T \text{ and } \bar{T}) \geq \text{threshold}\}$. Varying the threshold yields mixed networks with a varying number of interconnections. In the extreme case $mix(T, \bar{T}) = \emptyset$ (high threshold), $\mathcal{M}$ simply sums the outputs of $\mathcal{N}$ and $\bar{\mathcal{N}}$. As the threshold decreases interconnections between hidden layers are added – starting from hidden layer 2, then including hidden layer 4, and so on. The intuition from our analysis (sec. 5) is that additional interconnections result in a larger ensemble of hybrid mode trees, which in turn boosts the expressive power of the mixed network $\mathcal{M}$. As fig. 5 shows, this intuition indeed complies with the results in practice – classification accuracy improves as we increase the number of interconnections, without any additional cost in terms of computation or model capacity.[10]

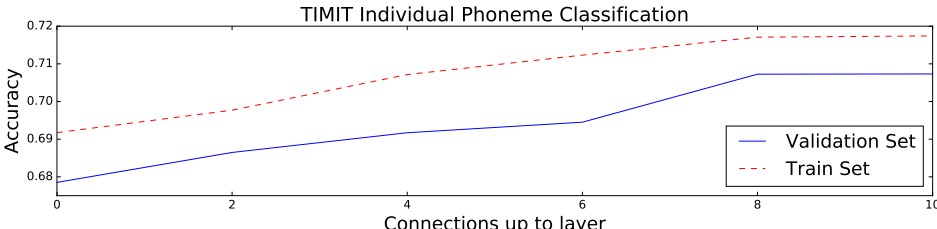

Figure 5: Experimental results – increasing the number of interconnections between hidden layers of different dilated convolutional networks improves accuracy, with no additional cost in computation or model capacity.

It is important to stress that our objective in the experiment was to evaluate, in the most controlled setting possible, the exact models covered by our analysis. We did not compare to state of the art results, as all phoneme recognition rates reported in the literature deviate from our basic setting – they heavily rely on data pre-processing (*e.g.* Mel-Frequency Cepstral Coefficients), prediction post-processing (*e.g.* Conditional Random Fields), or both. The recent DeepLab model (Chen et al. (2016)) has demonstrated that when combined with other techniques, mixing dilated convolutions can lead to state of the art image segmentation performance. We are currently pursuing similar results in the context of sequence processing tasks.

To conclude this section, we briefly convey implementation details behind the experiment. TIMIT dataset is an acoustic-phonetic corpus comprising 6300 sentences manually labeled at the phoneme level. We split the data into train and validation sets in accordance with Halberstadt (1998), and as advised by Lee and Hon (1989), mapped the 61 possible phoneme labels into 39 plus an additional "garbage" label. The task was then to classify individual phonemes into one of the latter categories. In accordance with WaveNet, the baseline dilated convolutional network had ReLU activation ($g(a, b) = \max\{a+b, 0\}$ – see sec. 3.1),[11] 32 channels per layer, and input vectors of dimension 256 holding one-hot quantizations of the audio signal. The number of layers $L$ was set to 12, corresponding to an input window of $N = 2^L = 4096$ samples, spanning 250ms of audio signal – standard practice with TIMIT dataset. The framework chosen for running the experiment was Caffe toolbox (Jia et al. (2014)), and we used Adam optimizer (Kingma and Ba (2014)) for training (with default hyper-parameters: moment decay rates $\beta_1 = 0.9, \beta_2 = 0.999$; learning rate $\alpha = 0.001$). Weight decay and batch size were set to $10^{-5}$ and 128 respectively. Models were trained for 35000 iterations, with learning rate decreased by a factor of 10 after 80% of iterations took place.

## 7 CONCLUSION

Nearly all state of the art deep networks these days (*e.g.* Szegedy et al. (2015); He et al. (2015); Huang et al. (2016b;a)) deviate from the simple feed-forward (chain) approach, employing various connectivity schemes between their layers. In this paper we studied the representational implications of connectivity in the context of dilated convolutional networks, a family of deep models delivering state of the art performance in audio and text processing tasks, underlying Google's WaveNet (van den Oord et al. (2016)) and ByteNet (Kalchbrenner et al. (2016)). We formulated our study through the notion of expressive efficiency, which refers to a situation where one network must grow unfeasibly large to realize (or approximate) functions of another. Our analysis shows that interconnecting hidden layers of different dilated convolutional networks can bring forth a model that is expressively efficient w.r.t. the individual networks it comprises. In particular, we show that a single connection between hidden layers can already lead to an almost quadratic gap, which in large-scale settings typically makes the difference between a model that is practical and one that is not. We empirically evaluate the analyzed networks, and find that the expressive efficiency brought forth by interconnectivity coincides with improved accuracies.

To date, formal analyses studying expressive efficiency have focused on the architectural feature of depth, showing instances where deep networks are expressively efficient w.r.t. shallow ones. These studies were motivated by the vast empirical evidence supporting the importance of depth. Our work thus provides a second exemplar of an architectural feature for which expressive efficiency and superior accuracies go hand in hand. This leads us to believe that expressive efficiency may serve a key role in the development of new tools for deep network design.

ACKNOWLEDGMENTS

This work was supported by Intel grant ICRI-CI #9-2012-6133, by ISF Center grant 1790/12, and by the European Research Council (TheoryDL project). Nadav Cohen was supported by a Google Doctoral Fellowship in Machine Learning.

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

NOTES

1  For every $(l, j)$, there exists a group of tensors $\{\phi^{l,j,\gamma}\}_\gamma$, formed through combinations of tensors from its "child" groups $\{\phi^{l-1,2j-1,\gamma}\}_\gamma$ and $\{\phi^{l-1,2j,\gamma}\}_\gamma$.

2  Binary mode trees lead to decompositions (eq. 3) that correspond to networks with size-2 convolutions. We limit ourselves to this case for simplicity of presentation. Our formulation can easily be extended to account for convolutions of arbitrary size by considering mode trees that are not necessarily binary, and by modifying the decomposition in eq. 3 to take (generalized) tensor products between an arbitrary number of tensors (not necessarily two).

3  A full binary tree is a tree in which all interior (non-leaf) nodes have exactly two children.

4  In general the number of tensors in the group $\{\phi^{\nu,\gamma}\}_\gamma$ may vary across nodes $\nu$, but for simplicity of presentation we assume that all groups comprise exactly $r$ tensors.

5  A perfect binary tree is a tree in which all interior (non-leaf) nodes have exactly two children and all leaves have exactly the same depth.

6  It is important to stress that not all choices of mode trees lead to networks resembling ones used in practice. For example, if different leaves in a tree have different depths, different inputs in the corresponding network pass through a different number of layers. Conversely, not every type of dilated convolutional network used in practice corresponds to a mode tree – only ones in which an input is connected to the output through a single path.

7  A few remarks are in order at this point:

   • The number of channels in each layer of $\mathcal{N}$ or $\bar{\mathcal{N}}$ corresponds to the constant $r$ in the respective tree decomposition (eq. 3 with underlying mode tree $T$ or $\bar{T}$ respectively). Similarly, the number of channels in each layer of each interconnected network in $\mathcal{M}$ corresponds to $r$ in the respective mixed decomposition (eq. 4). In both the tree and mixed decompositions, $r$, referred to as the *size constant*, stands for the number of tensors $\{\phi^{\nu,\gamma}\}_\gamma$ (respectively $\{\phi^{\bar{\nu},\gamma}\}_\gamma$) held in each node $\nu$ (respectively $\bar{\nu}$). We set this number uniformly across nodes, corresponding to uniformly sized layers across networks, merely for simplicity of presentation. Our formulations and analysis can easily be adapted to account for varying layer sizes, by allowing different nodes in a decomposition to hold a different number of tensors. Note that an implication of our uniform setting is that a network's input and output dimensions vary along with the size of its hidden layers. When replicating a function realized by a network using a larger network, we simply pad input vectors with zeros, and ignore the excess output coordinates.

   • An additional simplification we made relates to weight sharing. In both the tree and mixed decompositions, each interior node $\nu$ (respectively $\bar{\nu}$) has a separate set of weights $\{\mathbf{a}^{\nu,\gamma,\mathrm{I}}, \mathbf{a}^{\nu,\gamma,\mathrm{II}}\}_\gamma$ (respectively $\{\bar{\mathbf{a}}^{\bar{\nu},\gamma,\mathrm{I}}, \bar{\mathbf{a}}^{\bar{\nu},\gamma,\mathrm{II}}\}_\gamma$). This implies that in the corresponding networks, convolution filters may vary through time, *i.e.* different weights may be used against different portions of a convolved sequence. The more commonplace setting of stationary filters (standard convolutions) is obtained by restricting different nodes in a decomposition to possess the same weights. We do not introduce such restrictions into our formulations, as they make little difference in terms of the analysis, but on the other hand significantly burden presentation.

8  In accordance with the remark given at the beginning of this section, when using the (larger) mixed decomposition, we pad discretizers with zeros, and ignore the excess output tensors.

9  Treatment of additional cases can be achieved by deriving a result analogous to theorem 1, *i.e.* by characterizing matricization ranks brought forth by a tree decomposition (eq. 3) whose underlying operator $g(\cdot)$ corresponds to the architecture of interest (*e.g.* $g(a, b) = \max\{a + b, 0\}$ for networks with ReLU activation). Results along this line were established in Cohen and Shashua (2016). We defer their adoption and development to future work.

10  We note that in addition to the mixed dilated convolutional network $\mathcal{M}$, we also evaluated the individual networks $\mathcal{N}$ and $\bar{\mathcal{N}}$ – both reached accuracies comparable to $\mathcal{M}$ in the case of zero interconnections (output summation only).

11  The case of convolutional arithmetic circuits ($g(a, b) = a \cdot b$) was also evaluated, leading to the exact same trends as those observed with ReLU (fig. 5).

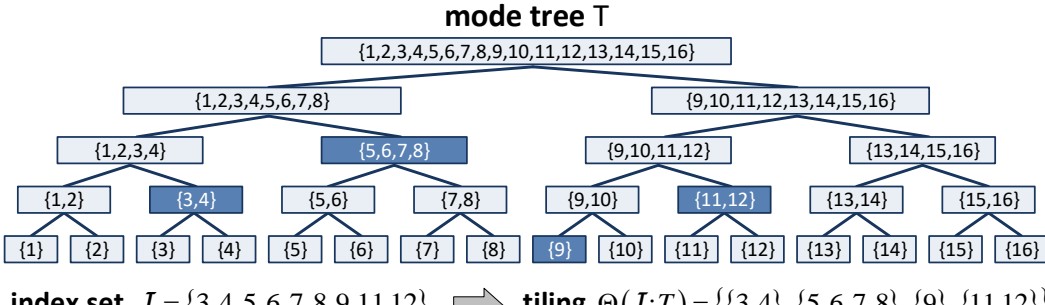

**index set** $\mathcal{I} = \{3,4,5,6,7,8,9,11,12\}$ ⟹ **tiling** $\Theta(\mathcal{I};T) = \{\{3,4\},\{5,6,7,8\},\{9\},\{11,12\}\}$

Figure 6: Mode tree $T$ along with a specific index set $\mathcal{I}$ and the resulting tiling $\Theta(\mathcal{I};T)$ (def. 3).

## A    PRELIMINARIES

The constructions and analyses delivered in this paper rely on concepts from the field of tensor analysis. Below we provide the minimal background required in order to follow our arguments.[1]

The core concept in tensor analysis is a *tensor*, which for our purposes may simply be thought of as a multi-dimensional array. The *order* of a tensor is defined to be the number of indexing entries in the array, which are referred to as *modes*. The *dimension* of a tensor in a particular mode is defined as the number of values that may be taken by the index in that mode. For example, a 4-by-3 matrix is a tensor of order 2, *i.e.* it has two modes, with dimension 4 in mode 1 and dimension 3 in mode 2. If $\mathcal{A}$ is a tensor of order $N$ and dimension $M_i$ in each mode $i \in \{1,\ldots,N\}$, the space of all configurations it can take is denoted, quite naturally, by $\mathbb{R}^{M_1 \times \cdots \times M_N}$.

A fundamental operator in tensor analysis is the *tensor product* (also known as *outer product*), which we denote by $\otimes$. It is an operator that intakes two tensors $\mathcal{A} \in \mathbb{R}^{M_1 \times \cdots \times M_P}$ and $\mathcal{B} \in \mathbb{R}^{M_{P+1} \times \cdots \times M_{P+Q}}$ (orders $P$ and $Q$ respectively), and returns a tensor $\mathcal{A} \otimes \mathcal{B} \in \mathbb{R}^{M_1 \times \cdots \times M_{P+Q}}$ (order $P+Q$) defined by: $(\mathcal{A} \otimes \mathcal{B})_{d_1 \ldots d_{P+Q}} = \mathcal{A}_{d_1 \ldots d_P} \cdot \mathcal{B}_{d_{P+1} \ldots d_{P+Q}}$. In Cohen and Shashua (2016) a generalization of the tensor product is defined, by replacing multiplication with a general operator $g(\cdot)$. Specifically, for a function $g : \mathbb{R} \times \mathbb{R} \to \mathbb{R}$ that is commutative ($g(a,b) = g(b,a)$ for all $a,b \in \mathbb{R}$), the *generalized tensor product*, denoted $\otimes_g$, is defined to be the operator that for input tensors $\mathcal{A} \in \mathbb{R}^{M_1 \times \cdots \times M_P}$ and $\mathcal{B} \in \mathbb{R}^{M_{P+1} \times \cdots \times M_{P+Q}}$ (orders $P$ and $Q$ respectively), returns the tensor $\mathcal{A} \otimes_g \mathcal{B} \in \mathbb{R}^{M_1 \times \cdots \times M_{P+Q}}$ (order $P+Q$) given by: $(\mathcal{A} \otimes_g \mathcal{B})_{d_1 \ldots d_{P+Q}} = g(\mathcal{A}_{d_1 \ldots d_P}, \mathcal{B}_{d_{P+1} \ldots d_{P+Q}})$.

An additional operator we make use of is *mode permutation*. Let $\mathcal{A}$ be a tensor of order $N$, and let $\sigma(\cdot)$ be a permutation over $N$ (bijective mapping from $\{1,\ldots,N\}$ to itself). The mode permutation of $\mathcal{A}$ w.r.t. $\sigma(\cdot)$, which by a slight abuse of notation is denoted $\sigma(\mathcal{A})$, is the order-$N$ tensor defined by: $\sigma(\mathcal{A})_{d_1 \ldots d_N} = \mathcal{A}_{d_{\sigma(1)} \ldots d_{\sigma(N)}}$. In words, $\sigma(\mathcal{A})$ is the tensor obtained by rearranging the modes of $\mathcal{A}$ in accordance with $\sigma(\cdot)$.

When studying tensors, it is oftentimes useful to arrange them as matrices, a procedure referred to as *matricization*. Let $\mathcal{A}$ be a tensor of order $N$ and dimension $M_i$ in each mode $i \in \{1,\ldots,N\}$, and let $\mathcal{I} \subset \{1,\ldots,N\}$ be a set of mode indexes, whose complement $\{1,\ldots,N\} \setminus \mathcal{I}$ we denote by $\mathcal{I}^c$. We may write $\mathcal{I} = \{i_1,\ldots,i_{|\mathcal{I}|}\}$ where $i_1 < \cdots < i_{|\mathcal{I}|}$, and similarly $\mathcal{I}^c = \{j_1,\ldots,j_{|\mathcal{I}^c|}\}$ where $j_1 < \cdots < j_{|\mathcal{I}^c|}$. The matricization of $\mathcal{A}$ w.r.t. $\mathcal{I}$, denoted $[\![\mathcal{A}]\!]_{\mathcal{I}}$, is the $\prod_{t=1}^{|\mathcal{I}|} M_{i_t}$-by-$\prod_{t=1}^{|\mathcal{I}^c|} M_{j_t}$ matrix holding the entries of $\mathcal{A}$ such that $\mathcal{A}_{d_1 \ldots d_N}$ is placed in row index $1 + \sum_{t=1}^{|\mathcal{I}|}(d_{i_t} - 1)\prod_{t'=t+1}^{|\mathcal{I}|} M_{i_{t'}}$ and column index $1 + \sum_{t=1}^{|\mathcal{I}^c|}(d_{j_t} - 1)\prod_{t'=t+1}^{|\mathcal{I}^c|} M_{j_{t'}}$. If $\mathcal{I} = \emptyset$ or $\mathcal{I} = \{1,\ldots,N\}$, then by definition $[\![\mathcal{A}]\!]_{\mathcal{I}}$ is a row or column (respectively) vector of dimension $\prod_{t=1}^{N} M_t$ holding $\mathcal{A}_{d_1 \ldots d_N}$ in entry $1 + \sum_{t=1}^{N}(d_t - 1)\prod_{t'=t+1}^{N} M_{t'}$.

To conclude this appendix, we hereinafter list notational conventions used throughout the paper. We denote tensors with uppercase calligraphic letters, *e.g.* $\mathcal{A}$, and in some cases, with the Greek letters $\phi$, $\varphi$ or $\psi$. Subscripts are used to refer to individual tensor entries, *e.g.* $\mathcal{A}_{d_1 \ldots d_N} \in \mathbb{R}$, whereas

---

[1] The viewpoint we adopt is actually a concrete special case of a more abstract algebraic viewpoint of tensor analysis, as presented for example in Hackbusch (2012). We limit ourselves to this concrete viewpoint since it suffices for our needs and is easier to grasp.

superscripts indicate the location of a tensor in some annotated collection, for example $\mathcal{A}^y$ stands for the $y$'th tensor in the collection $\mathcal{A}^1 \ldots \mathcal{A}^r$. Vectors are typically denoted with boldface lowercase letters, *e.g.* $\mathbf{a}$, where again subscripts refer to an individual entry (*e.g.* $a_\alpha \in \mathbb{R}$), and superscripts to the identity of a vector within some annotated collection (*e.g.* $\mathbf{a}^{l,j}$ is the $(l,j)$'th vector in the set $\{\mathbf{a}^{l,j}\}_{l=1\ldots L,j=1\ldots N}$). We use non-boldface lowercase or uppercase letters (*e.g.* $l$ or $L$) to denote scalars, and in this case, both subscripts and superscripts distinguish between objects in an annotated set (*e.g.* $l_i, l^i, L_i, L^i \in \mathbb{R}$). Finally, for a positive integer $N \in \mathbb{N}$, we use $[N]$ as shorthand for the set $\{1, \ldots, N\}$.

## B DETAILED DESCRIPTION OF THE BASELINE ARCHITECTURE

In this appendix we describe in detail the architecture of the baseline dilated convolutional network (fig. 1). The input to the network is a sequence of vectors $(\mathbf{x}[t])_t \subset \mathbb{R}^{r_0}$, where $t$ is a natural time index. A size-2 convolutional layer with dilation-1, *i.e.* with contiguous filters, maps this input into the hidden sequence $(\mathbf{h}^{(1)}[t])_t \subset \mathbb{R}^{r_1}$. Specifically, entry $\gamma \in [r_1]$ of $\mathbf{h}^{(1)}[t]$ is obtained by applying the filter formed by $\mathbf{a}^{1,\gamma,\mathrm{I}}, \mathbf{a}^{1,\gamma,\mathrm{II}} \in \mathbb{R}^{r_0}$ to time points $t-1, t$ of the input: $h^{(1)}[t]_\gamma = g(\langle \mathbf{a}^{1,\gamma,\mathrm{I}}, \mathbf{x}[t-1] \rangle, \langle \mathbf{a}^{1,\gamma,\mathrm{II}}, \mathbf{x}[t] \rangle)$. We use $g(\cdot)$ here to denote the binary function combining two size-1 convolutions into a single size-2 convolution with non-linearity. Different choices of $g(\cdot)$ lead to different convolutional operators, for example $g(a, b) := \max\{a+b, 0\}$ leads to standard convolution followed by rectified linear activation (*ReLU*, Nair and Hinton (2010)), whereas $g(a, b) = a \cdot b$ gives rise to what is known as a *convolutional arithmetic circuit* (Cohen et al. (2016b)). Following the first hidden layer, $L-1$ size-2 convolutional layers with increasing dilations are applied. Specifically, for $l = 2, \ldots, L-1$, hidden layer $l$ maps the sequence $(\mathbf{h}^{(l-1)}[t])_t \subset \mathbb{R}^{r_{l-1}}$ into $(\mathbf{h}^{(l)}[t])_t \subset \mathbb{R}^{r_l}$ using filters with dilation-$2^{l-1}$, *i.e.* with an internal temporal gap of $2^{l-1}-1$ points: $h^{(l)}[t]_\gamma = g(\langle \mathbf{a}^{l,\gamma,\mathrm{I}}, \mathbf{h}^{(l-1)}[t-2^{l-1}] \rangle, \langle \mathbf{a}^{l,\gamma,\mathrm{II}}, \mathbf{h}^{(l-1)}[t] \rangle)$. The last convolutional layer maps $(\mathbf{h}^{(L-1)}[t])_t$ into network output sequence $(\mathbf{o}[t])_t \subset \mathbb{R}^{r_L}$ using filters with dilation-$2^{L-1}$: $o[t]_y = g(\langle \mathbf{a}^{L,y,\mathrm{I}}, \mathbf{h}^{(L-1)}[t-2^{L-1}] \rangle, \langle \mathbf{a}^{L,y,\mathrm{II}}, \mathbf{h}^{(L-1)}[t] \rangle)$.

Altogether, the architectural parameters of the network are the number of convolutional layers $L$, the convolutional operator $g(\cdot)$, the input dimension $r_0$, the number of channels $r_l$ for each hidden layer $l \in [L-1]$, and the output dimension $r_L$. The learnable parameters are the convolution weights $\mathbf{a}^{l,\gamma,\mathrm{I}}, \mathbf{a}^{l,\gamma,\mathrm{II}} \in \mathbb{R}^{r_{l-1}}$ for channel $\gamma \in [r_l]$ of layer $l \in [L]$.

## C DERIVATION OF THE BASELINE DECOMPOSITION

In this appendix we derive the baseline decomposition (eq. 2) – a parameterization of grid tensors (eq. 1) discretizing input-output mappings of the baseline dilated convolutional network (fig. 1; app. B). As discussed in sec. 3.1, $\mathbf{o}[t]$ – the network output at time $t$, is a function of $\mathbf{x}[t\text{-}N+1] \ldots \mathbf{x}[t]$ – its input over the last $N := 2^L$ time points. We would like to show that for any $d_1 \ldots d_N \in [M]$, entry $(d_1, \ldots, d_N)$ of a tensor $\mathcal{A}^y$ generated by eq. 2, is equal to coordinate $y$ of network output $\mathbf{o}[t]$ under the following input assignment: $\mathbf{x}[t\text{-}N+1] = \mathbf{v}^{(d_1)}, \ldots, \mathbf{x}[t] = \mathbf{v}^{(d_N)}$. To achieve this, we prove by induction that under the latter assignment, for every $l \in [L] \cup \{0\}$, $j \in [N/2^l]$ and $\gamma \in [r_l]$, coordinate $\gamma$ of the network's depth-$l$ sequence (input $(\mathbf{x}[t])_t$ for $l = 0$; hidden sequence $(\mathbf{h}^{(l)}[t])_t$ for $l \in [L-1]$; output $(\mathbf{o}[t])_t$ for $l = L$) at time $t - N + j \cdot 2^l$, is equal to entry $(d_{(j-1)2^l+1}, \ldots, d_{(j-1)2^l+2^l})$ of the tensor $\phi^{l,j,\gamma}$ in the baseline decomposition (eq. 2). The desired result then follows from the case $l = L, j = 1, \gamma = y$.

When $l = 0$, the inductive hypothesis is trivial – coordinate $\gamma$ of the input sequence at time $t - N + j$, *i.e.* $x[t - N + j]_\gamma$, is by definition of our assignment equal to $v_\gamma^{(d_j)}$ – entry $d_j$ of the tensor $\phi^{0,j,\gamma}$ (see eq. 2). Assume now that the inductive hypothesis holds whenever $l = k$, and consider the tensor $\phi^{k+1,j,\gamma}$ for some $j \in [N/2^{k+1}]$ and $\gamma \in [r_{k+1}]$. From the baseline decomposition (eq. 2):

$$\phi^{k+1,j,\gamma} = \left( \sum_{\alpha=1}^{r_k} a_\alpha^{k+1,\gamma,\mathrm{I}} \cdot \phi^{k,2j-1,\alpha} \right) \otimes_g \left( \sum_{\alpha=1}^{r_k} a_\alpha^{k+1,\gamma,\mathrm{II}} \cdot \phi^{k,2j,\alpha} \right)$$

Focusing on entry $(d_{(j-1)2^{k+1}+1}, \ldots, d_{(j-1)2^{k+1}+2^{k+1}})$ of the left-hand side, while recalling the definition of the generalized tensor product $\otimes_g$ (app. A), we may write:

$$\phi_{d_{(j-1)2^{k+1}+1}, \ldots, d_{(j-1)2^{k+1}+2^{k+1}}}^{k+1,j,\gamma} =$$

$$g \left( \sum_{\alpha=1}^{r_k} a_\alpha^{k+1,\gamma,\mathrm{I}} \cdot \phi_{d_{(2j-2)2^k+1}, \ldots, d_{(2j-2)2^k+2^k}}^{k,2j-1,\alpha}, \sum_{\alpha=1}^{r_k} a_\alpha^{k+1,\gamma,\mathrm{II}} \cdot \phi_{d_{(2j-1)2^k+1}, \ldots, d_{(2j-1)2^k+2^k}}^{k,2j,\alpha} \right) \quad (5)$$

By our inductive assumption:

$$
\begin{array}{rcll}
\phi^{k,2j-1,\alpha}_{d_{(2j-2)2^k+1},\ldots,d_{(2j-2)2^k+2^k}} & = & h^{(k)}[t-N+(2j-1)\cdot 2^k]_\alpha & \forall\alpha\in[r_k] \\[2mm]
\phi^{k,2j,\alpha}_{d_{(2j-1)2^k+1},\ldots,d_{(2j-1)2^k+2^k}} & = & h^{(k)}[t-N+2j\cdot 2^k]_\alpha & \forall\alpha\in[r_k]
\end{array}
$$

where we overload notation in the case $k=0$, letting $(\mathbf{h}^{(0)}[t])_t$ stand for the input sequence $(\mathbf{x}[t])_t$. Plugging the latter into eq. 5, we obtain:

$$
\phi^{k+1,j,\gamma}_{d_{(j-1)2^{k+1}+1},\ldots,d_{(j-1)2^{k+1}+2^{k+1}}} =
$$
$$
g\left(\left\langle\mathbf{a}^{k+1,\gamma,\mathrm{I}},\mathbf{h}^{(k)}[t-N+(2j-1)\cdot 2^k]\right\rangle,\left\langle\mathbf{a}^{k+1,\gamma,\mathrm{II}},\mathbf{h}^{(k)}[t-N+2j\cdot 2^k]\right\rangle\right)
$$

By the definition of the baseline dilated convolutional network (fig. 1; app. B), the latter expression is precisely equal to coordinate $\gamma$ of the sequence $(\mathbf{h}^{(k+1)}[t])_t$ (or $(\mathbf{o}[t])_t$ if $k=L-1$) at time $t-N+j\cdot 2^{k+1}$. This proves that our inductive hypothesis holds when $l=k+1$, and in general.

## D  ANNOTATIONS OF THE TREE AND MIXED DECOMPOSITIONS

In this appendix we describe the tree and mixed decompositions (eq. 3 and 4 respectively), whose annotations were omitted from the text due to lack of space.

Let $T$ be a binary mode tree over $[N]$ (def. 1). The tree decomposition of $T$ (eq. 3) iteratively assigns a group of ($2^{|\nu|}$-order) tensors $\{\phi^{\nu,\gamma}\}_{\gamma\in[r]}$ for each node $\nu$ in $T$, based on weight vectors $\{\mathbf{a}^{\nu,\gamma,\mathrm{I}},\mathbf{a}^{\nu,\gamma,\mathrm{II}}\in\mathbb{R}^r\}_{\gamma\in[r]}$ defined for each interior node $\nu\in int(T)$. Specifically, the decomposition traverses through $T$ in a depth-first fashion, and for each node $\nu$, assigns the tensor group $\{\phi^{\nu,\gamma}\}_\gamma$ as follows:

- If $\nu$ is a leaf, *i.e.* $\nu=\{j\}$ for some $j\in[N]$, its tensors ($\{\phi^{\nu,\gamma}\}_\gamma$) are set directly by the discretizers $\mathbf{v}^{(1)}\ldots\mathbf{v}^{(M)}$ ($v^{(i)}_\gamma$ in eq. 3 stands for coordinate $\gamma$ of $\mathbf{v}^{(i)}$).

- If $\nu$ is an interior node, *i.e.* $\nu\in int(T)$, its tensors are set through combinations of the tensors of its children ($\{\phi^{C_\mathrm{I}(\nu;T),\gamma}\}_\gamma$ and $\{\phi^{C_\mathrm{II}(\nu;T),\gamma}\}_\gamma$). These combinations are based on the weight vectors $\{\mathbf{a}^{\nu,\gamma,\mathrm{I}}\}_\gamma$ and $\{\mathbf{a}^{\nu,\gamma,\mathrm{II}}\}_\gamma$, as depicted in eq. 3 ($a^{\nu,\gamma,\mathrm{I}}_\alpha$ and $a^{\nu,\gamma,\mathrm{II}}_\alpha$ there stand for coordinate $\alpha$ of $\mathbf{a}^{\nu,\gamma,\mathrm{I}}$ and $\mathbf{a}^{\nu,\gamma,\mathrm{II}}$ respectively). The permutation $\sigma^{(\nu;T)}(\cdot)$ in the assignment of $\phi^{\nu,\gamma}$ arranges the modes of the tensor (see app. A) such that these comply with a sorted ordering of $\nu$. Namely, if we denote by $i_1<\cdots<i_{|C_\mathrm{I}(\nu;T)|}$ the elements of $C_\mathrm{I}(\nu;T)\subset[N]$, and by $j_1<\cdots<j_{|C_\mathrm{II}(\nu;T)|}$ the elements of $C_\mathrm{II}(\nu;T)\subset[N]$, the permutation $\sigma^{(\nu;T)}:[2^{|\nu|}]\to[2^{|\nu|}]$ is the one that sorts the tuple $(i_1,\ldots,i_{|C_\mathrm{I}(\nu;T)|},j_1,\ldots,j_{|C_\mathrm{II}(\nu;T)|})$ in ascending order.

The final outcome of the tree decomposition, *i.e.* the generated grid tensors $\{\mathcal{A}^y\}_y$, are precisely the tensor group $\{\phi^{[N],\gamma}\}_\gamma$ corresponding to the root of $T$ ($[N]$).

Heading on to the mixed decomposition (eq. 4), let $\bar{T}$ be an additional binary mode tree over $[N]$ (def. 1). When considering the tree decomposition of $\bar{T}$, we use $\{\bar{\phi}^{\bar{\nu},\gamma}\}_{\gamma\in[r]}$ to denote the tensor group of node $\bar{\nu}\in\bar{T}$, and $\{\bar{\mathbf{a}}^{\bar{\nu},\gamma,\mathrm{I}},\bar{\mathbf{a}}^{\bar{\nu},\gamma,\mathrm{II}}\}_{\gamma\in[r]}$ to denote the weights of interior node $\bar{\nu}\in int(\bar{T})$. For a chosen collection of mixture nodes $mix(T,\bar{T})\subset int(T)\cap int(\bar{T})$, the mixed decomposition of $T$ and $\bar{T}$ blends together their tree decompositions by running these in parallel, while exchanging tensors whenever a mixture node is reached. The procedure is formulated in eq. 4 – annotation follows:

- As in the basic tree decomposition (eq. 3), the first step (lines 1-2) is to assign tensors corresponding to the leaf nodes ($\{1\}\ldots\{N\}$) via discretizers $\mathbf{v}^{(1)}\ldots\mathbf{v}^{(M)}$.

- The outer loop in line 3 traverses $\mu$ through mixture nodes and the root node in inclusion order, *i.e.* such that a node (subset of $[N]$) is always reached after all nodes strictly contained in it.

- Lines 4-5 (respectively 6-7) are the same as in the tree decomposition (eq. 3), except that instead of running through the entire interior of $T$ (respectively $\bar{T}$), they cover a segment of it. This segment continues where the previous ones left off, and comprises only nodes (subsets of $[N]$) contained in $\mu$ (including $\mu$ itself).

- Line 8 is where the mixing takes place – here half the tensors corresponding to node $\mu$ in the decomposition of $T$ ($\{\phi^{\mu,\gamma}\}_\gamma$), are exchanged for half the tensors corresponding to $\mu$ in the decomposition of $\bar{T}$ ($\{\bar{\phi}^{\mu,\gamma}\}_\gamma$).

- Finally, after $\mu$ has reached the root node $[N]$ and the decompositions of $T$ and $\bar{T}$ have concluded, line 9 sums the output tensors of these decompositions ($\{\phi^{[N],y}\}_y$ and $\{\bar{\phi}^{[N],y}\}_y$ respectively), producing the grid tensors $\{\mathcal{A}^y\}_y$.

## E  DEFERRED PROOFS

### E.1  PROOF OF CLAIM 1

We initiate the proof by introducing notations that will allow a more compact presentation. Hereinafter, we let $\{\mathbf{a}^{H,\nu,\gamma,\mathrm{I}}, \mathbf{a}^{H,\nu,\gamma,\mathrm{II}} \in \mathbb{R}^{r_{mix}/2}\}_{\nu \in int(H), \gamma \in [r_{mix}/2]}$ stand for the weights in the tree decomposition of the hybrid mode tree $H$ (eq. 3 with size constant $r = r_{mix}/2$ and underlying mode tree given by def. 2). Similarly, we use $\{\mathbf{a}^{T,\nu,\gamma,\mathrm{I}}, \mathbf{a}^{T,\nu,\gamma,\mathrm{II}} \in \mathbb{R}^{r_{mix}}\}_{\nu \in int(T), \gamma \in [r_{mix}]}$ and $\{\mathbf{a}^{\bar{T},\nu,\gamma,\mathrm{I}}, \mathbf{a}^{\bar{T},\nu,\gamma,\mathrm{II}} \in \mathbb{R}^{r_{mix}}\}_{\nu \in int(\bar{T}), \gamma \in [r_{mix}]}$ to denote the weights, corresponding to $T$ and $\bar{T}$ (respectively), in the mixed decomposition (eq. 4 with size constant $r = r_{mix}$). Recall that by construction (def. 2), $int(H)$ – the interior of $H$, consists of different segments (collections of nodes), each taken from either $int(T)$ or $int(\bar{T})$. We define $t : int(H) \to \{T, \bar{T}\}$ to be the function indicating which tree an interior node in $H$ came from. Specifically, if the node $\nu \in int(H)$ originated from $T$ we have $t(\nu) = T$, and on the other hand, if its source is $\bar{T}$ then $t(\nu) = \bar{T}$. By convention, feeding $t(\cdot)$ with an argument outside $int(H)$ yields something that is different from both $T$ and $\bar{T}$. For example, if $\nu \in int(H)$ is the root node, *i.e.* $\nu = [N]$, then $P(\nu; H)$ – its parent in $H$, is undefined and we have $t(P(\nu; H)) \neq t(\nu)$. Similarly, if the child $C_{\mathrm{I}}(\nu; H)$ of $\nu \in int(H)$ is a leaf, it is outside the domain of $t(\cdot)$ and thus $t(\nu) \neq t(C_{\mathrm{I}}(\nu; H))$.

Given a setting of weights $\{\mathbf{a}^{H,\nu,\gamma,\mathrm{I}}, \mathbf{a}^{H,\nu,\gamma,\mathrm{II}}\}_{\nu,\gamma}$ for the tree decomposition of $H$, we would like to show that there exists a setting of weights $\{\mathbf{a}^{T,\nu,\gamma,\mathrm{I}}, \mathbf{a}^{T,\nu,\gamma,\mathrm{II}}\}_{\nu,\gamma}$ and $\{\mathbf{a}^{\bar{T},\nu,\gamma,\mathrm{I}}, \mathbf{a}^{\bar{T},\nu,\gamma,\mathrm{II}}\}_{\nu,\gamma}$ for the mixed decomposition of $T$ and $\bar{T}$, such that the latter generates grid tensors identical to those of the former. More precisely, for any collection of discretizers $\{\mathbf{v}^{(i)} \in \mathbb{R}^{r_{mix}/2}\}_{i \in [M]}$ fed into the tree decomposition of $H$, leading the latter to produce grid tensors $\{\mathcal{A}^y\}_{y \in [r_{mix}/2]}$, we would like the mixed decomposition to be such that when fed with the padded discretizers $\{[(\mathbf{v}^{(i)})^\top \ \mathbf{0}]^\top \in \mathbb{R}^{r_{mix}}\}_{i \in [M]}$, the first $r_{mix}/2$ grid tensors it generates are equal to $\{\mathcal{A}^y\}_{y \in [r_{mix}/2]}$. We prove existence of the sought after weight setting constructively, by presenting an explicit procedure for assigning $\{\mathbf{a}^{T,\nu,\gamma,\mathrm{I}}, \mathbf{a}^{T,\nu,\gamma,\mathrm{II}}\}_{\nu,\gamma}$ and $\{\mathbf{a}^{\bar{T},\nu,\gamma,\mathrm{I}}, \mathbf{a}^{\bar{T},\nu,\gamma,\mathrm{II}}\}_{\nu,\gamma}$ based on $\{\mathbf{a}^{H,\nu,\gamma,\mathrm{I}}, \mathbf{a}^{H,\nu,\gamma,\mathrm{II}}\}_{\nu,\gamma}$:

Initialize:

$$\mathbf{a}^{T,\nu,\gamma,\mathrm{I}} = \mathbf{a}^{T,\nu,\gamma,\mathrm{II}} = \mathbf{0} \quad \forall \nu \in int(T), \gamma \in [r_{mix}]$$

$$\mathbf{a}^{\bar{T},\nu,\gamma,\mathrm{I}} = \mathbf{a}^{\bar{T},\nu,\gamma,\mathrm{II}} = \mathbf{0} \quad \forall \nu \in int(\bar{T}), \gamma \in [r_{mix}]$$

For $\nu$ in $int(H)$ (depth-first order):

$$\mathbf{a}^{t(\nu),\nu,\gamma+\frac{1}{2}r_{mix},\mathrm{I}} = \begin{cases} \left[\mathbf{0}^\top \ (\mathbf{a}^{H,\nu,\gamma,\mathrm{I}})^\top\right]^\top & , t(\nu) = t(C_{\mathrm{I}}(\nu; H)) \\ \left[(\mathbf{a}^{H,\nu,\gamma,\mathrm{I}})^\top \ \mathbf{0}^\top\right]^\top & , t(\nu) \neq t(C_{\mathrm{I}}(\nu; H)) \end{cases} \quad \forall \gamma \in [r_{mix}/2]$$

$$\mathbf{a}^{t(\nu),\nu,\gamma+\frac{1}{2}r_{mix},\mathrm{II}} = \begin{cases} \left[\mathbf{0}^\top \ (\mathbf{a}^{H,\nu,\gamma,\mathrm{II}})^\top\right]^\top & , t(\nu) = t(C_{\mathrm{II}}(\nu; H)) \\ \left[(\mathbf{a}^{H,\nu,\gamma,\mathrm{II}})^\top \ \mathbf{0}^\top\right]^\top & , t(\nu) \neq t(C_{\mathrm{II}}(\nu; H)) \end{cases} \quad \forall \gamma \in [r_{mix}/2]$$

If $t(P(\nu; H)) \neq t(\nu)$ :

Swap $\mathbf{a}^{t(\nu),\nu,\gamma,\mathrm{I}} \longleftrightarrow \mathbf{a}^{t(\nu),\nu,\gamma+\frac{1}{2}r_{mix},\mathrm{I}} \quad \forall \gamma \in [r_{mix}/2]$

Swap $\mathbf{a}^{t(\nu),\nu,\gamma,\mathrm{II}} \longleftrightarrow \mathbf{a}^{t(\nu),\nu,\gamma+\frac{1}{2}r_{mix},\mathrm{II}} \quad \forall \gamma \in [r_{mix}/2]$ (6)

The idea behind this assignment is as follows. The computation corresponding to a node in the tree decomposition of $H$, is carried out, in the mixed decomposition of $T$ and $\bar{T}$, by the respective node in the respective source tree. That is to say, the computation of $\nu \in int(H)$ in the tree decomposition is carried out by $\nu \in int(t(\nu))$ in the mixed decomposition. $\nu \in int(t(\nu))$ uses half ($r_{mix}/2$) of its

weight vectors, and in each used weight vector, half ($r_{mix}/2$) of the coordinates hold actual (non-zero) values – a copy of the respective weight from $\nu \in int(H)$. The choice of which weight vectors to use, and which coordinates to use in the active weight vectors, depends on the tree-transitioning scheme. If the parent of $\nu$ in $H$ came from the same tree as $\nu$, *i.e.* $t(P(\nu;H)) = t(\nu)$, $\nu \in int(t(\nu))$ in the mixed decomposition uses weight vectors with higher indexes ($\gamma \in r_{mix}/2 + [r_{mix}/2]$), as these relate to tensors that are not exchanged (see eq. 4). On the other hand, if $t(P(\nu;H)) \neq t(\nu)$, weight vectors with lower indexes ($\gamma \in [r_{mix}/2]$) are used, so that the computations (tensors) will be sent to the opposite tree. The analogous rationale holds for the children of $\nu$ in $H$ ($C_{\mathrm{I}}(\nu;H)$ and $C_{\mathrm{II}}(\nu;H)$). If a child came from the same tree as $\nu$, upper coordinates of the appropriate weight vectors are used, so that computations (tensors) coming from the present tree are collected. On the other hand, if the child came from the opposite tree, lower coordinates are used and computations (tensors) from that tree are fetched. Altogether, the assignment in eq. 6 meets our requirements, and thus concludes the proof.

$\square$

### E.2 PROOF OF THEOREM 1

#### E.2.1 SKETCH

The proof proceeds in three stages. In the first stage we matricize the tree decomposition of $T$, *i.e.* transform it from a tensor decomposition generating $\{\mathcal{A}^y\}_y$ to a matrix decomposition generating $\{[\![\mathcal{A}^y]\!]_{\mathcal{I}}\}_y$. In this transformation, instances of the tensor product $\otimes$ ($\otimes_g$ with $g(a,b) = a \cdot b$ – see app. A) convert to a *Kronecker product* $\odot$. The second stage of the proof establishes the upper bound stated in the theorem, by showing that for each $y$, $[\![\mathcal{A}^y]\!]_{\mathcal{I}}$ is equal to a product of matrices, one of which has size $r^{|\Theta(\mathcal{I};T)|}$-by-$r^{|\Theta(\mathcal{I}^c;T)|}$. The key idea in this stage is the propagation of elements out of the matrix decomposition, using the relation $(AA') \odot (BB') = (A \odot A')(B \odot B')$. The third and final stage of the proof establishes the lower bound stated in the theorem. Here again, elements are propagated out of the matrix decomposition, allowing the construction of a concrete configuration of weights ($\{\mathbf{a}^{\nu,\gamma,\mathrm{I}}, \mathbf{a}^{\nu,\gamma,\mathrm{II}}\}_{\nu,\gamma}$) for which the lower bound holds. The fact that the lower bound holds almost always is then a direct corollary of app. G, where it is shown that the tree decomposition admits maximal matricization ranks almost always when $g(\cdot)$ is the product operator.

The study of matricization ranks under hierarchical tensor decompositions is of significant interest, particularly in the context of deep learning. Cohen et al. (2016b) proved the lower bound in the theorem for the specific case where $T$ is the mode tree corresponding to the baseline dilated convolutional network (see fig. 2(a)), and $\mathcal{I} = \{1, 3, \ldots, N-1\}$. The result was used to establish exponential expressive efficiency of deep convolutional arithmetic circuits w.r.t. shallow ones. Cohen and Shashua (2017) later extended the analysis by deriving upper bounds for arbitrary index sets $\mathcal{I}$, using them to study the ability of deep convolutional arithmetic circuits to model correlations among regions of their input. The bounds used in Cohen and Shashua (2017) were loose, and in fact trivial for many choices of index sets $\mathcal{I}$. We here treat arbitrary mode trees $T$ and index sets $\mathcal{I}$, proving upper and lower bounds that are tight, oftentimes exact. Such tight bounds are necessary for identifying expressive efficiency that is not exponential, as we do in this paper. The key to deriving the bounds is the aforementioned idea of propagating elements out of a matrix decomposition.

#### E.2.2 COMPLETE PROOF

Since we are dealing with a single particular mode tree $T$, we omit it from our notations throughout the proof. Specifically, we denote by $C_{\mathrm{I}}(\nu)$ and $C_{\mathrm{II}}(\nu)$ (instead of $C_{\mathrm{I}}(\nu;T)$ and $C_{\mathrm{II}}(\nu;T)$) the children of an interior node $\nu \in int(T)$; by $\Theta(\mathcal{I})$ and $\Theta(\mathcal{I}^c)$ (instead of $\Theta(\mathcal{I};T)$ and $\Theta(\mathcal{I}^c;T)$) the tilings of $\mathcal{I}$ and $\mathcal{I}^c$ (respectively) w.r.t. $T$ (see def. 3); and by $\sigma^{(\nu)}(\cdot)$ (instead of $\sigma^{(\nu;T)}(\cdot)$) the permutation corresponding to $\nu \in int(T)$ in the tree decomposition (eq. 3).

The first stage of the proof is to derive a matricized form of the tree decomposition, shedding light into the manner in which grid tensor matricizations $\{[\![\mathcal{A}^y]\!]_{\mathcal{I}}\}_y$ are generated. As a preparatory step in this direction, we define the notion of an *index set reduction*. Let $\nu \subset [N]$ be a node in $T$, whose elements we denote by $i_1 < \cdots < i_{|\nu|}$. The reduction of $\mathcal{I}$ onto $\nu$ is defined as follows:

$$\mathcal{I}|_\nu := \{j \in [|\nu|] : i_j \in \mathcal{I} \cap \nu\} \tag{7}$$

In words, it is the set of indexes corresponding to the intersection $\mathcal{I} \cap \nu$ inside $\nu$. Besides index set reduction, an additional tool we will be using is the *Kronecker product* – a matrix operator we denote

by $\odot$. For two matrices $A \in \mathbb{R}^{M_1 \times M_2}$ and $B \in \mathbb{R}^{N_1 \times N_2}$, $A \odot B$ is the matrix in $\mathbb{R}^{M_1 N_1 \times M_2 N_2}$ holding $A_{ij} B_{kl}$ in row index $(i-1)N_1 + k$ and column index $(j-1)N_2 + l$.

Consider the central relation in the tree decomposition (eq. 3), while noticing that $\otimes_g \equiv \otimes$ in our setting ($g(\cdot)$ is the product operator – see app. A):

$$\underbrace{\phi^{\nu,\gamma}}_{\text{order } 2^{|\nu|}} = \sigma^{(\nu)}\left(\left(\sum_{\alpha=1}^{r} a_{\alpha}^{\nu,\gamma,\mathrm{I}} \cdot \phi^{C_{\mathrm{I}}(\nu),\alpha}\right) \otimes \left(\sum_{\alpha=1}^{r} a_{\alpha}^{\nu,\gamma,\mathrm{II}} \cdot \phi^{C_{\mathrm{II}}(\nu),\alpha}\right)\right) \tag{8}$$

Suppose we would like to matricize the tensor $\phi^{\nu,\gamma}$ w.r.t. the reduction $\mathcal{I}|_\nu$. If all elements of $C_{\mathrm{I}}(\nu)$ were smaller than those of $C_{\mathrm{II}}(\nu)$, the permutation $\sigma^{(\nu)}(\cdot)$ would be the identity (see sec. 3.2), and the following matrix relation would hold:

$$
\begin{aligned}
[\![\phi^{\nu,\gamma}]\!]_{\mathcal{I}|_\nu} &= \left[\!\left[\left(\sum_{\alpha=1}^{r} a_{\alpha}^{\nu,\gamma,\mathrm{I}} \cdot \phi^{C_{\mathrm{I}}(\nu),\alpha}\right) \otimes \left(\sum_{\alpha=1}^{r} a_{\alpha}^{\nu,\gamma,\mathrm{II}} \cdot \phi^{C_{\mathrm{II}}(\nu),\alpha}\right)\right]\!\right]_{\mathcal{I}|_\nu} \\
&= \left[\!\left[\sum_{\alpha=1}^{r} a_{\alpha}^{\nu,\gamma,\mathrm{I}} \cdot \phi^{C_{\mathrm{I}}(\nu),\alpha}\right]\!\right]_{\mathcal{I}|_{C_{\mathrm{I}}(\nu)}} \odot \left[\!\left[\sum_{\alpha=1}^{r} a_{\alpha}^{\nu,\gamma,\mathrm{II}} \cdot \phi^{C_{\mathrm{II}}(\nu),\alpha}\right]\!\right]_{\mathcal{I}|_{C_{\mathrm{II}}(\nu)}} \\
&= \left(\sum_{\alpha=1}^{r} a_{\alpha}^{\nu,\gamma,\mathrm{I}} \cdot [\![\phi^{C_{\mathrm{I}}(\nu),\alpha}]\!]_{\mathcal{I}|_{C_{\mathrm{I}}(\nu)}}\right) \odot \left(\sum_{\alpha=1}^{r} a_{\alpha}^{\nu,\gamma,\mathrm{II}} \cdot [\![\phi^{C_{\mathrm{II}}(\nu),\alpha}]\!]_{\mathcal{I}|_{C_{\mathrm{II}}(\nu)}}\right)
\end{aligned}
$$

In general however, elements in $C_{\mathrm{I}}(\nu)$ could be greater than ones in $C_{\mathrm{II}}(\nu)$, and so eq. 8 includes a tensor mode sorting via $\sigma^{(\nu)}(\cdot)$. In matricized form, this amounts to rearranging rows and columns through appropriate permutation matrices $Q^{(\nu)}$ and $\bar{Q}^{(\nu)}$ respectively:

$$[\![\phi^{\nu,\gamma}]\!]_{\mathcal{I}|_\nu} = Q^{(\nu)}\left(\left(\sum_{\alpha=1}^{r} a_{\alpha}^{\nu,\gamma,\mathrm{I}} \cdot [\![\phi^{C_{\mathrm{I}}(\nu),\alpha}]\!]_{\mathcal{I}|_{C_{\mathrm{I}}(\nu)}}\right) \odot \left(\sum_{\alpha=1}^{r} a_{\alpha}^{\nu,\gamma,\mathrm{II}} \cdot [\![\phi^{C_{\mathrm{II}}(\nu),\alpha}]\!]_{\mathcal{I}|_{C_{\mathrm{II}}(\nu)}}\right)\right) \bar{Q}^{(\nu)}$$

We thus arrive at the following matrix form of eq. 3, referred to as the *matricized tree decomposition*:

For $j = 1 \ldots N$:
$$[\![\phi^{\{j\},\gamma}]\!]_{\mathcal{I}|_{\{j\}}} = \left[\!\left[[v_\gamma^{(1)}, \ldots, v_\gamma^{(M)}]^\top\right]\!\right]_{\mathcal{I}|_{\{j\}}} \quad \forall \gamma \in [r]$$

For $\nu$ in $int(T)$ (depth-first order):
$$[\![\phi^{\nu,\gamma}]\!]_{\mathcal{I}|_\nu} = Q^{(\nu)}\left(\left(\sum_{\alpha=1}^{r} a_{\alpha}^{\nu,\gamma,\mathrm{I}}[\![\phi^{C_{\mathrm{I}}(\nu),\alpha}]\!]_{\mathcal{I}|_{C_{\mathrm{I}}(\nu)}}\right) \odot \left(\sum_{\alpha=1}^{r} a_{\alpha}^{\nu,\gamma,\mathrm{II}}[\![\phi^{C_{\mathrm{II}}(\nu),\alpha}]\!]_{\mathcal{I}|_{C_{\mathrm{II}}(\nu)}}\right)\right) \bar{Q}^{(\nu)} \quad \forall \gamma \in [r]$$

$$[\![\mathcal{A}^y]\!]_{\mathcal{I}} = [\![\phi^{[N],y}]\!]_{\mathcal{I}|_{[N]}} \quad \forall y \in [r] \tag{9}$$

Next, we move on to the second stage of the proof, where we establish the upper bound stated in the theorem:

$$rank[\![\mathcal{A}^y]\!]_{\mathcal{I}} \leq r^{\min\{|\Theta(\mathcal{I})|,|\Theta(\mathcal{I}^c)|\}} \quad \forall y \tag{10}$$

We begin by "propagating outwards" the permutation matrices $Q^{([N])}$ and $\bar{Q}^{([N])}$ corresponding to the root node $[N]$ in the matricized tree decomposition (eq. 9). Namely, for every $\gamma \in [r]$, we replace the matrix $[\![\phi^{[N],\gamma}]\!]_{\mathcal{I}|_{[N]}}$ by:

$$B^{[N],\gamma} := \left(\sum_{\alpha=1}^{r} a_{\alpha}^{[N],\gamma,\mathrm{I}}[\![\phi^{C_{\mathrm{I}}([N]),\alpha}]\!]_{\mathcal{I}|_{C_{\mathrm{I}}([N])}}\right) \odot \left(\sum_{\alpha=1}^{r} a_{\alpha}^{[N],\gamma,\mathrm{II}}[\![\phi^{C_{\mathrm{II}}([N]),\alpha}]\!]_{\mathcal{I}|_{C_{\mathrm{II}}([N])}}\right)$$

and accordingly move $Q^{([N])}$ and $\bar{Q}^{([N])}$ to the assignments of $\{[\![\mathcal{A}^y]\!]_{\mathcal{I}}\}_y$. This gives rise to the following decomposition:

For $j = 1 \ldots N$:

$$[\![\phi^{\{j\},\gamma}]\!]_{\mathcal{I}|_{\{j\}}} = \left[\!\left[[v_\gamma^{(1)}, \ldots, v_\gamma^{(M)}]^\top\right]\!\right]_{\mathcal{I}|_{\{j\}}} \quad \forall \gamma \in [r]$$

For $\nu$ in $int(T) \setminus \{[N]\}$ (depth-first order):

$$[\![\phi^{\nu,\gamma}]\!]_{\mathcal{I}|_\nu} = Q^{(\nu)}\left(\left(\sum_{\alpha=1}^r a_\alpha^{\nu,\gamma,\mathrm{I}}[\![\phi^{C_\mathrm{I}(\nu),\alpha}]\!]_{\mathcal{I}|_{C_\mathrm{I}(\nu)}}\right) \odot \left(\sum_{\alpha=1}^r a_\alpha^{\nu,\gamma,\mathrm{II}}[\![\phi^{C_\mathrm{II}(\nu),\alpha}]\!]_{\mathcal{I}|_{C_\mathrm{II}(\nu)}}\right)\right)\bar{Q}^{(\nu)} \quad \forall \gamma \in [r]$$

$$B^{[N],\gamma} = \left(\sum_{\alpha=1}^r a_\alpha^{[N],\gamma,\mathrm{I}}[\![\phi^{C_\mathrm{I}([N]),\alpha}]\!]_{\mathcal{I}|_{C_\mathrm{I}([N])}}\right) \odot \left(\sum_{\alpha=1}^r a_\alpha^{[N],\gamma,\mathrm{II}}[\![\phi^{C_\mathrm{II}([N]),\alpha}]\!]_{\mathcal{I}|_{C_\mathrm{II}([N])}}\right) \quad \forall \gamma \in [r]$$

$$[\![\mathcal{A}^y]\!]_{\mathcal{I}} = Q^{([N])}B^{[N],y}\bar{Q}^{([N])} \quad \forall y \in [r]$$

Consider now $C_\mathrm{I}([N])$ – a child of the root node $[N]$, and suppose we would like to similarly propagate outwards its permutation matrices $Q^{(C_\mathrm{I}([N]))}$ and $\bar{Q}^{(C_\mathrm{I}([N]))}$. We may define, for every $\gamma \in [r]$:

$$B^{C_\mathrm{I}([N]),\gamma} := \left(\sum_{\alpha=1}^r a_\alpha^{C_\mathrm{I}([N]),\gamma,\mathrm{I}}[\![\phi^{C_\mathrm{I}(C_\mathrm{I}([N])),\alpha}]\!]_{\mathcal{I}|_{C_\mathrm{I}(C_\mathrm{I}([N]))}}\right) \odot \left(\sum_{\alpha=1}^r a_\alpha^{C_\mathrm{I}([N]),\gamma,\mathrm{II}}[\![\phi^{C_\mathrm{II}(C_\mathrm{I}([N])),\alpha}]\!]_{\mathcal{I}|_{C_\mathrm{II}(C_\mathrm{I}([N]))}}\right)$$

which in turn implies:

$$\begin{aligned}
B^{[N],\gamma} &= \left(\sum_{\alpha=1}^r a_\alpha^{[N],\gamma,\mathrm{I}}Q^{(C_\mathrm{I}([N]))}B^{C_\mathrm{I}([N]),\alpha}\bar{Q}^{(C_\mathrm{I}([N]))}\right) \odot \left(\sum_{\alpha=1}^r a_\alpha^{[N],\gamma,\mathrm{II}}[\![\phi^{C_\mathrm{II}([N]),\alpha}]\!]_{\mathcal{I}|_{C_\mathrm{II}([N])}}\right) \\
&= \left(Q^{(C_\mathrm{I}([N]))}\left(\sum_{\alpha=1}^r a_\alpha^{[N],\gamma,\mathrm{I}}B^{C_\mathrm{I}([N]),\alpha}\right)\bar{Q}^{(C_\mathrm{I}([N]))}\right) \odot \left(\sum_{\alpha=1}^r a_\alpha^{[N],\gamma,\mathrm{II}}[\![\phi^{C_\mathrm{II}([N]),\alpha}]\!]_{\mathcal{I}|_{C_\mathrm{II}([N])}}\right)
\end{aligned}$$

Now, for any matrices $A, A', B, B'$ such that $AA'$ and $BB'$ are defined, the following equality holds: $(AA') \odot (BB') = (A \odot A')(B \odot B')$ (see Bellman (1970) for proof). We may therefore write:

$$B^{[N],\gamma} =$$
$$\left(Q^{(C_\mathrm{I}([N]))} \odot I\right)\left(\left(\sum_{\alpha=1}^r a_\alpha^{[N],\gamma,\mathrm{I}}B^{C_\mathrm{I}([N]),\alpha}\right) \odot \left(\sum_{\alpha=1}^r a_\alpha^{[N],\gamma,\mathrm{II}}[\![\phi^{C_\mathrm{II}([N]),\alpha}]\!]_{\mathcal{I}|_{C_\mathrm{II}([N])}}\right)\right)\left(\bar{Q}^{(C_\mathrm{I}([N]))} \odot \bar{I}\right)$$

where $I$ and $\bar{I}$ are identity matrices of appropriate sizes. Propagating outwards the matrices $Q^{(C_\mathrm{I}([N]))} \odot I$ and $\bar{Q}^{(C_\mathrm{I}([N]))} \odot \bar{I}$ (while redefining $B^{[N],\gamma}$ appropriately), we arrive at the following decomposition:

For $j = 1 \ldots N$:

$$[\![\phi^{\{j\},\gamma}]\!]_{\mathcal{I}|_{\{j\}}} = \left[\!\left[[v_\gamma^{(1)}, \ldots, v_\gamma^{(M)}]^\top\right]\!\right]_{\mathcal{I}|_{\{j\}}} \quad \forall \gamma \in [r]$$

For $\nu$ in $int(T) \setminus \{[N], C_\mathrm{I}([N])\}$ (depth-first order):

$$[\![\phi^{\nu,\gamma}]\!]_{\mathcal{I}|_\nu} = Q^{(\nu)}\left(\left(\sum_{\alpha=1}^r a_\alpha^{\nu,\gamma,\mathrm{I}}[\![\phi^{C_\mathrm{I}(\nu),\alpha}]\!]_{\mathcal{I}|_{C_\mathrm{I}(\nu)}}\right) \odot \left(\sum_{\alpha=1}^r a_\alpha^{\nu,\gamma,\mathrm{II}}[\![\phi^{C_\mathrm{II}(\nu),\alpha}]\!]_{\mathcal{I}|_{C_\mathrm{II}(\nu)}}\right)\right)\bar{Q}^{(\nu)} \quad \forall \gamma \in [r]$$

$$B^{C_\mathrm{I}([N]),\gamma} = \left(\sum_{\alpha=1}^r a_\alpha^{C_\mathrm{I}([N]),\gamma,\mathrm{I}}[\![\phi^{C_\mathrm{I}(C_\mathrm{I}([N])),\alpha}]\!]_{\mathcal{I}|_{C_\mathrm{I}(C_\mathrm{I}([N]))}}\right)$$
$$\odot \left(\sum_{\alpha=1}^r a_\alpha^{C_\mathrm{I}([N]),\gamma,\mathrm{II}}[\![\phi^{C_\mathrm{II}(C_\mathrm{I}([N])),\alpha}]\!]_{\mathcal{I}|_{C_\mathrm{II}(C_\mathrm{I}([N]))}}\right) \quad \forall \gamma \in [r]$$

$$B^{[N],\gamma} = \left(\sum_{\alpha=1}^r a_\alpha^{[N],\gamma,\mathrm{I}}B^{C_\mathrm{I}([N]),\alpha}\right) \odot \left(\sum_{\alpha=1}^r a_\alpha^{[N],\gamma,\mathrm{II}}[\![\phi^{C_\mathrm{II}([N]),\alpha}]\!]_{\mathcal{I}|_{C_\mathrm{II}([N])}}\right) \quad \forall \gamma \in [r]$$

$$[\![\mathcal{A}^y]\!]_{\mathcal{I}} = \left(Q^{([N])}(Q^{(C_\mathrm{I}([N]))} \odot I)\right)B^{[N],y}\left((\bar{Q}^{(C_\mathrm{I}([N]))} \odot \bar{I})\bar{Q}^{([N])}\right) \quad \forall y \in [r]$$

Continuing this process, we propagate outwards the permutation matrices $Q^{(\nu)}$ and $\bar{Q}^{(\nu)}$ of all nodes $\nu$ in the tree that are not members of the tilings $\Theta(\mathcal{I})$ or $\Theta(\mathcal{I}^c)$ (see def. 3), and are not descendants of such. This brings forth the following decomposition:

For $j = 1 \ldots N$:
$$[\![\phi^{\{j\},\gamma}]\!]_{\mathcal{I}|_{\{j\}}} = \left[\!\left[ [v_\gamma^{(1)}, \ldots, v_\gamma^{(M)}]^\top \right]\!\right]_{\mathcal{I}|_{\{j\}}} \quad \forall \gamma \in [r]$$

For $\nu$ in $int(T) \cap \{\text{nodes in } \Theta(\mathcal{I}) \text{ or } \Theta(\mathcal{I}^c) \text{ or descendants of such}\}$ (depth-first order):
$$[\![\phi^{\nu,\gamma}]\!]_{\mathcal{I}|_\nu} = Q^{(\nu)} \left( \left( \sum_{\alpha=1}^r a_\alpha^{\nu,\gamma,\mathrm{I}} [\![\phi^{C_\mathrm{I}(\nu),\alpha}]\!]_{\mathcal{I}|_{C_\mathrm{I}(\nu)}} \right) \odot \left( \sum_{\alpha=1}^r a_\alpha^{\nu,\gamma,\mathrm{II}} [\![\phi^{C_\mathrm{II}(\nu),\alpha}]\!]_{\mathcal{I}|_{C_\mathrm{II}(\nu)}} \right) \right) \bar{Q}^{(\nu)} \quad \forall \gamma \in [r]$$

For $\nu$ in $\Theta(\mathcal{I}) \cup \Theta(\mathcal{I}^c)$:
$$B^{\nu,\gamma} = [\![\phi^{\nu,\gamma}]\!]_{\mathcal{I}|_\nu} \quad \forall \gamma \in [r]$$

For $\nu$ in $int(T) \backslash \{\text{nodes in } \Theta(\mathcal{I}) \text{ or } \Theta(\mathcal{I}^c) \text{ or descendants of such}\}$ (depth-first order):
$$B^{\nu,\gamma} = \left( \sum_{\alpha=1}^r a_\alpha^{\nu,\gamma,\mathrm{I}} B^{C_\mathrm{I}(\nu),\alpha} \right) \odot \left( \sum_{\alpha=1}^r a_\alpha^{\nu,\gamma,\mathrm{II}} B^{C_\mathrm{II}(\nu),\alpha} \right) \quad \forall \gamma \in [r]$$

$$[\![\mathcal{A}^y]\!]_{\mathcal{I}} = A \cdot B^{[N],y} \cdot \bar{A} \quad \forall y \in [r], \text{ for appropriate matrices } A \text{ and } \bar{A}$$

Consider now a node $\nu \in int(T)$ whose child belongs to a tiling – without loss of generality $C_\mathrm{I}(\nu)$ belongs to $\Theta(\mathcal{I})$. Notice that in this case $B^{C_\mathrm{I}(\nu),\alpha}$ is a column vector for every $\alpha \in [r]$. We may thus define $B^{C_\mathrm{I}(\nu)}$ to be the matrix whose $\alpha$'th column is $B^{C_\mathrm{I}(\nu),\alpha}$, and get the following equalities:

$$B^{\nu,\gamma} = \left( B^{C_\mathrm{I}(\nu)} \mathbf{a}^{\nu,\gamma,\mathrm{I}} \right) \odot \left( \sum_{\alpha=1}^r a_\alpha^{\nu,\gamma,\mathrm{II}} B^{C_\mathrm{II}(\nu),\alpha} \right) = \left( B^{C_\mathrm{I}(\nu)} \odot I \right) \left( \mathbf{a}^{\nu,\gamma,\mathrm{I}} \odot \sum_{\alpha=1}^r a_\alpha^{\nu,\gamma,\mathrm{II}} B^{C_\mathrm{II}(\nu),\alpha} \right)$$

where again, $I$ is an appropriately sized identity matrix. This implies that we can propagate outwards $B^{C_\mathrm{I}(\nu)} \odot I$, just as we have done with permutation matrices. Applying this procedure to all nodes in the tilings $\Theta(\mathcal{I})$ and $\Theta(\mathcal{I}^c)$, we arrive at the decomposition below:

For $\nu$ in $\Theta(\mathcal{I})$:
$$B^{\nu,\gamma} = \mathbf{e}^{(\gamma)} \quad \forall \gamma \in [r]$$

For $\nu$ in $\Theta(\mathcal{I}^c)$:
$$B^{\nu,\gamma} = (\mathbf{e}^{(\gamma)})^\top \quad \forall \gamma \in [r]$$

For $\nu$ in $int(T) \backslash \{\text{nodes in } \Theta(\mathcal{I}) \text{ or } \Theta(\mathcal{I}^c) \text{ or descendants of such}\}$ (depth-first order):
$$B^{\nu,\gamma} = \left( \sum_{\alpha=1}^r a_\alpha^{\nu,\gamma,\mathrm{I}} B^{C_\mathrm{I}(\nu),\alpha} \right) \odot \left( \sum_{\alpha=1}^r a_\alpha^{\nu,\gamma,\mathrm{II}} B^{C_\mathrm{II}(\nu),\alpha} \right) \quad \forall \gamma \in [r]$$

$$[\![\mathcal{A}^y]\!]_{\mathcal{I}} = A \cdot B^{[N],y} \cdot \bar{A} \quad \forall y \in [r], \text{ for appropriate matrices } A \text{ and } \bar{A}$$

Notice that for compactness in writing we made use of the fact that $\mathbf{a}^{\nu,\gamma,\mathrm{I}} = \sum_{\alpha=1}^r a_\alpha^{\nu,\gamma,\mathrm{II}} \mathbf{e}^{(\alpha)}$, where $\mathbf{e}^{(\alpha)}$, $\alpha \in [r]$, is the vector in $\mathbb{R}^r$ holding 1 in entry $\alpha$ and 0 in the rest. Note also that in this decomposition, as opposed to the previous ones, the matrices $A$ and $\bar{A}$ are not global constants that depend only on $T$. Rather, they also depend on $[\![\phi^{\nu,\gamma}]\!]_{\mathcal{I}|_\nu}$ for tiling nodes $\nu \in \Theta(\mathcal{I}) \cup \Theta(\mathcal{I}^c)$, and thus are ultimately determined through a hidden computation that is not specified above. This hidden computation is outside our scope, as we are only interested in the size of the matrices $\{B^{[N],y}\}_y$. It is not difficult to see that this size is precisely $r^{|\Theta(\mathcal{I})|}$-by-$r^{|\Theta(\mathcal{I}^c)|}$, meaning that the ranks of $\{B^{[N],y}\}_y$ are no more than $r^{\min\{|\Theta(\mathcal{I})|,|\Theta(\mathcal{I}^c)|\}}$. Since these ranks are greater than or equal to those of $\{[\![\mathcal{A}^y]\!]_{\mathcal{I}}\}_y$, the sought after upper bound (eq. 10) indeed holds.

In the third and final stage of the proof, we establish the lower bound stated in the theorem, namely, that for all configurations of weights $\{\mathbf{a}^{\nu,\gamma,\mathrm{I}}, \mathbf{a}^{\nu,\gamma,\mathrm{II}}\}_{\nu,\gamma}$ but a set of Lebesgue measure zero:

$$rank[\![\mathcal{A}^y]\!]_{\mathcal{I}} \geq r^{|\{(\nu_1,\nu_2)\in\Theta(\mathcal{I})\times\Theta(\mathcal{I}^c): \nu_1 \text{ and } \nu_2 \text{ are siblings in } T \text{ with depth}>1\}|} \quad \forall y \qquad (11)$$

We reduce the problem in three successive steps:

- A tree decomposition (eq. 3) with a product operator $g(\cdot)$ admits maximal matricization ranks almost always (see app. G). Therefore, to prove that eq. 11 holds for all weight settings but a set of Lebesgue measure zero, it suffices to find a particular weight setting for which the inequality holds.

- By assumption, the discretizers $\{\mathbf{v}^{(i)}\}_{i\in[M]}$ span $\mathbb{R}^r$. Without loss of generality, assume that $\{\mathbf{v}^{(i)}\}_{i\in[r]}$ are linearly independent, and consider the sub-tensors of $\{\mathcal{A}^y\}_y$ formed by restricting their indexes to the range $1\ldots r$ (instead of $1\ldots M$). The matricizations of these sub-tensors w.r.t. $\mathcal{I}$ are sub-matrices of $\{[\![\mathcal{A}^y]\!]_{\mathcal{I}}\}_y$, thus any lower bound on ranks of the former matricizations immediately translates to a lower bound on ranks of the latter. Since the sub-tensors are precisely the grid tensors that would have been generated by the tree decomposition (eq. 3) had we omitted the trailing discretizers $\{\mathbf{v}^{(i)}\}_{i\in[M]\setminus[r]}$, establishing eq. 11 in the case $M=r$ proves that it holds in general ($M\geq r$).

- Bearing in mind that we assume $M=r$ (and linear independence of $\{\mathbf{v}^{(i)}\}_{i\in[r]}$), denote by $V$ the $r$-by-$r$ matrix holding $\mathbf{v}^{(i)}$ in its $i$'th row, *i.e.* $V:=[\mathbf{v}^{(1)}\cdots\mathbf{v}^{(r)}]^\top$. From the tree decomposition (eq. 3) it is evident that the discretizers affect generated grid tensors only through products of the form $V\mathbf{a}^{\nu,\gamma\mathrm{I}}$ or $V\mathbf{a}^{\nu,\gamma\mathrm{II}}$, where $\nu$ is a parent of a leaf node in $T$. Since $V$ is invertible ($\{\mathbf{v}^{(i)}\}_{i\in[r]}$ are linearly independent), its exact value has no effect on the class of representable grid tensors – any change it undergoes may be accounted for by the weights $\mathbf{a}^{\nu,\gamma\mathrm{I}}$ and $\mathbf{a}^{\nu,\gamma\mathrm{II}}$ that multiply it (these weights do not appear elsewhere in the decomposition). Accordingly, for establishing a lower bound on achievable grid tensor matricization ranks, the value of $V$ is irrelevant (so long as it is invertible), and we may assume, without loss of generality, that $V$ is the identity matrix, *i.e.* that $\mathbf{v}^{(i)}=\mathbf{e}^{(i)}$ for all $i\in[r]$.

Taking into account the above reductions, our objective is to show that there exists a setting of weights $\{\mathbf{a}^{\nu,\gamma,\mathrm{I}},\mathbf{a}^{\nu,\gamma,\mathrm{II}}\}_{\nu,\gamma}$, such that the following special case of the matricized tree decomposition (eq. 9) generates matricizations meeting the lower bound in eq. 11:

For $j$ in $\mathcal{I}$:
$$[\![\phi^{\{j\},\gamma}]\!]_{\mathcal{I}|_{\{j\}}}=\mathbf{e}^{(\gamma)}\quad\forall\gamma\in[r]$$

For $j$ in $\mathcal{I}^c$:
$$[\![\phi^{\{j\},\gamma}]\!]_{\mathcal{I}|_{\{j\}}}=(\mathbf{e}^{(\gamma)})^\top\quad\forall\gamma\in[r]$$

For $\nu$ in $int(T)$ (depth-first order):
$$[\![\phi^{\nu,\gamma}]\!]_{\mathcal{I}|_\nu}=Q^{(\nu)}\left(\left(\sum_{\alpha=1}^r a_\alpha^{\nu,\gamma,\mathrm{I}}[\![\phi^{C_\mathrm{I}(\nu),\alpha}]\!]_{\mathcal{I}|_{C_\mathrm{I}(\nu)}}\right)\odot\left(\sum_{\alpha=1}^r a_\alpha^{\nu,\gamma,\mathrm{II}}[\![\phi^{C_\mathrm{II}(\nu),\alpha}]\!]_{\mathcal{I}|_{C_\mathrm{II}(\nu)}}\right)\right)\bar{Q}^{(\nu)}\ \forall\gamma\in[r]$$

$$[\![\mathcal{A}^y]\!]_{\mathcal{I}}=[\![\phi^{[N],y}]\!]_{\mathcal{I}|_{[N]}}\quad\forall y\in[r]$$

Similarly to the procedure carried out in the second stage of the proof (establishing the upper bound in eq. 10), we now propagate outwards the permutation matrices $Q^{(\nu)}$ and $\bar{Q}^{(\nu)}$ corresponding to all interior nodes $\nu\in int(T)$. This brings forth the following decomposition:

For $j$ in $\mathcal{I}$:
$$B^{\{j\},\gamma}=\mathbf{e}^{(\gamma)}\quad\forall\gamma\in[r]$$

For $j$ in $\mathcal{I}^c$:
$$B^{\{j\},\gamma}=(\mathbf{e}^{(\gamma)})^\top\quad\forall\gamma\in[r]$$

For $\nu$ in $int(T)$ (depth-first order):
$$B^{\nu,\gamma}=\left(\sum_{\alpha=1}^r a_\alpha^{\nu,\gamma,\mathrm{I}}B^{C_\mathrm{I}(\nu),\alpha}\right)\odot\left(\sum_{\alpha=1}^r a_\alpha^{\nu,\gamma,\mathrm{II}}B^{C_\mathrm{II}(\nu),\alpha}\right)\quad\forall\gamma\in[r]$$

$$[\![\mathcal{A}^y]\!]_{\mathcal{I}}=A\cdot B^{[N],y}\cdot\bar{A}\ \forall y\in[r],\ \text{for appropriate matrices } A \text{ and } \bar{A} \tag{12}$$

The matrices $A$ and $\bar{A}$ in the assignments of $\{[\![\mathcal{A}^y]\!]_{\mathcal{I}}\}_y$ essentially collect all permutation matrices $\{Q^{(\nu)}\}_\nu$ and $\{\bar{Q}^{(\nu)}\}_\nu$ (respectively) that have been propagated outwards. Specifically, $A$ (respectively $\bar{A}$) is a product of factors, each of the form $I\odot Q^{(\nu)}\odot I'$ (respectively $I\odot\bar{Q}^{(\nu)}I'$) for a

different interior node $\nu$ and appropriately sized identity matrices $I$ and $I'$. Since permutation matrices are invertible, and since the Kronecker product between two invertible matrices is invertible as well (see Bellman (1970) for proof), we conclude that the matrices $A$ and $\bar{A}$ are invertible. Therefore, for every $y \in [r]$, the rank of $[\![\mathcal{A}^y]\!]_{\mathcal{I}}$ is equal to that of $B^{[N],y}$. It thus suffices to find a setting of weights $\{\mathbf{a}^{\nu,\gamma,\mathrm{I}}, \mathbf{a}^{\nu,\gamma,\mathrm{II}}\}_{\nu,\gamma}$ for which:

$$rank(B^{[N],\gamma}) \geq r^{|\{(\nu_1,\nu_2)\in\Theta(\mathcal{I})\times\Theta(\mathcal{I}^c):\ \nu_1 \text{ and } \nu_2 \text{ are siblings in } T \text{ with depth}>1\}|} \quad \forall\gamma \in [r] \qquad (13)$$

Disregard the trivial case where there exist siblings $\nu_1 \in \Theta(\mathcal{I})$ and $\nu_2 \in \Theta(\mathcal{I}^c)$ of depth $1$,[2] and consider the following weight setting:

- $\nu$ is a node in $\Theta(\mathcal{I})$ or $\Theta(\mathcal{I}^c)$, or a descendant of such:

$$\mathbf{a}^{\nu,\gamma,\mathrm{I}} = \mathbf{a}^{\nu,\gamma,\mathrm{II}} = \mathbf{e}^{(\gamma)} \quad \forall\gamma \in [r]$$

- $\nu$ has one child in $\Theta(\mathcal{I})$ and the other in $\Theta(\mathcal{I}^c)$:

$$\mathbf{a}^{\nu,\gamma,\mathrm{I}} = \mathbf{a}^{\nu,\gamma,\mathrm{II}} = \mathbf{e}^{(\gamma)} \quad \forall\gamma \in [r]$$

- $\nu$ is the root node $[N]$:

$$\mathbf{a}^{\nu,\gamma,\mathrm{I}} = \mathbf{a}^{\nu,\gamma,\mathrm{II}} = \mathbf{e}^{(1)} \quad \forall\gamma \in [r]$$

- $\nu$ meets neither of the above ($\mathbf{0}$ and $\mathbf{1}$ here denote the all-zero and all-one vectors in $\mathbb{R}^r$, respectively):

$$\mathbf{a}^{\nu,1,\mathrm{I}} = \begin{cases} \mathbf{1} & , C_{\mathrm{I}}(\nu) \text{ has one child in } \Theta(\mathcal{I}) \text{ and the other in } \Theta(\mathcal{I}^c) \\ \mathbf{e}^{(1)} & , \text{otherwise} \end{cases}$$

$$\mathbf{a}^{\nu,1,\mathrm{II}} = \begin{cases} \mathbf{1} & , C_{\mathrm{II}}(\nu) \text{ has one child in } \Theta(\mathcal{I}) \text{ and the other in } \Theta(\mathcal{I}^c) \\ \mathbf{e}^{(1)} & , \text{otherwise} \end{cases}$$

$$\mathbf{a}^{\nu,\gamma,\mathrm{I}} = \mathbf{a}^{\nu,\gamma,\mathrm{II}} = \mathbf{0} \quad \forall\gamma \in [r] \setminus \{1\}$$

Plugging this into the decomposition in eq. 12, one readily sees that:

- For every $\nu \in \Theta(\mathcal{I})$, $\{B^{\nu,\gamma}\}_{\gamma\in[r]}$ are indicator column vectors (one entry holds 1, the rest hold 0) such that $B^{\nu,\gamma} \neq B^{\nu,\gamma'}$ if $\gamma \neq \gamma'$. The same holds for $\nu \in \Theta(\mathcal{I}^c)$, but with the vectors being rows.
- If $\nu$ has one child in $\Theta(\mathcal{I})$ and the other in $\Theta(\mathcal{I}^c)$, $\{B^{\nu,\gamma}\}_{\gamma\in[r]}$ are indicator matrices, where both the row and column indexes of the active entry do not repeat as $\gamma$ varies.
- The matrices $\{B^{[N],\gamma}\}_{\gamma\in[r]}$ corresponding to the root node $[N]$ are equal to one another, given by a joint Kronecker product between all of the following:
  - $B^{\nu,1}$ for every node $\nu$ in either $\Theta(\mathcal{I})$ or $\Theta(\mathcal{I}^c)$ which does not have a sibling in the other
  - $\sum_{\alpha=1}^{r} B^{\nu,\alpha}$ for every node $\nu$ that has one child in $\Theta(\mathcal{I})$ and the other in $\Theta(\mathcal{I}^c)$

According to the first observation above, $B^{\nu,1}$ has rank 1 for every $\nu$ in $\Theta(\mathcal{I})$ or $\Theta(\mathcal{I}^c)$. The second observation implies that $\sum_{\alpha=1}^{r} B^{\nu,\alpha}$ has rank $r$ for every node $\nu$ that has one child in $\Theta(\mathcal{I})$ and the other in $\Theta(\mathcal{I}^c)$. In turn, and while taking into account the rank-multiplicative property of the Kronecker product ($rank(A \odot A') = rank(A) \cdot rank(A')$ – see Bellman (1970) for proof), the third observation implies:

$$rank(B^{[N],\gamma}) = r^{|\{(\nu_1,\nu_2)\in\Theta(\mathcal{I})\times\Theta(\mathcal{I}^c):\ \nu_1 \text{ and } \nu_2 \text{ are siblings in } T\}|} \quad \forall\gamma \in [r]$$

We thus have found weights $\{\mathbf{a}^{\nu,\gamma,\mathrm{I}}, \mathbf{a}^{\nu,\gamma,\mathrm{II}}\}_{\nu,\gamma}$ for which eq. 13 holds.[3] This establishes the sought after lower bound on matricization ranks (eq. 11), completing the proof of the theorem.

$\square$

---

[2] In this case $\mathcal{I}$ and $\mathcal{I}^c$ are the children of the root node $[N]$, and the maximal rank of $B^{[N],\gamma}$ is 1 for every $\gamma \in [r]$.

[3] This applies to all but the trivial case where $\mathcal{I}$ is such that there exist siblings $\nu_1 \in \Theta(\mathcal{I})$ and $\nu_2 \in \Theta(\mathcal{I}^c)$ of depth 1 ($\mathcal{I}$ and $\mathcal{I}^c$ are the children of the root node $[N]$). In the latter case the lower bound in eq. 13 can be met trivially.

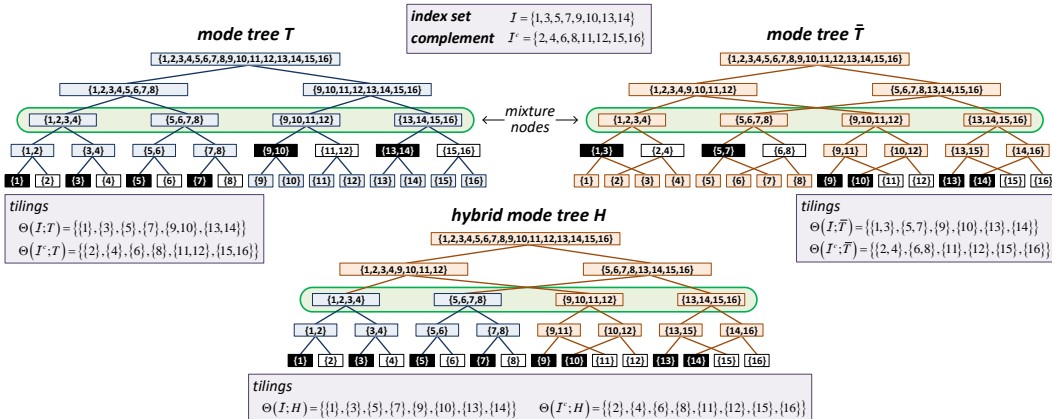

Figure 7: Best viewed in color. Two mode trees $T$ and $\bar{T}$ with a possible choice of mixture nodes (same as in fig. 3(a) and 4(a)), along with a particular formed hybrid tree $H$. An index set $\mathcal{I}$ and its complement $\mathcal{I}^c$ are tiled into more pieces by $H$ than they are by $T$ and $\bar{T}$, leading the former to generate grid tensors with higher matricization ranks (theorem 1).

## F  DEMONSTRATION OF EXPRESSIVE EFFICIENCY

In this appendix, omitted from the text due to lack of space, we demonstrate the application of theorem 1 for establishing expressive efficiency. In particular, we use the theorem to derive corollary 1.

Consider our exemplar mode trees illustrated in fig. 2. Specifically, let $T$ be the mode tree corresponding to the baseline dilated convolutional network (dilation $2^{l-1}$ in layer $l \in [L] = [\log_2 N]$ – see sec. 3.1), and let $\bar{T}$ be the mode tree corresponding to the network obtained by swapping dilations of even and odd layers (such that layer $l$ has dilation $2^{l-2}$ if $l$ is even, and $2^l$ if $l$ is odd). $T$ is a perfect binary tree whose depth-$l$ nodes, $l \in \{0, 1, \ldots, L\}$, are $(k-1)N/2^l + [N/2^l]$ for $k \in [2^l]$.[4] $\bar{T}$ is also perfect and has the same even-depth nodes, but its odd-depth nodes differ – they are generated by splitting parents into children holding non-contiguous quadrants. Suppose we choose $mix(T, \bar{T})$ to include the set of nodes in $T$ and $\bar{T}$ whose depth is $L-2$, and consider the hybrid mode tree $H$ formed by taking the segments (see def. 2) of the first half of these nodes from $T$, and the rest of the tree from $\bar{T}$. An illustration of $T$, $\bar{T}$ and $H$ in this setting, for the case $L = 4$, is given in fig. 7. Now, let the index set $\mathcal{I}$ consist of every second index in $[N/2]$, and every second pair of indexes in $N/2 + [N/2]$, i.e. $\mathcal{I} := \{2k-1 : k \in [N/4]\} \cup \{N/2 + 4k - k' : k \in [N/8], k' = 2, 3\}$. As illustrated in fig. 7, the mode tree $T$ tiles (see def. 3) the lower half of $\mathcal{I}$ into singletons, and its upper half into pairs. The same applies to $T$'s tiling of $\mathcal{I}$'s complement $\mathcal{I}^c := [N] \setminus \mathcal{I}$. Moreover, for every node in the tiling $\Theta(\mathcal{I}; T)$, there exists a sibling in $\Theta(\mathcal{I}^c; T)$ (and vice versa). By theorem 1, this implies that the tree decomposition of $T$ generates grid tensors whose matricizations w.r.t. $\mathcal{I}$ have rank $r^{N/4 + N/8}$. A similar situation occurs with the mode tree $\bar{T}$, under which $\mathcal{I}$ and $\mathcal{I}^c$ are tiled into pairs in their lower halves and into singletons in their top halves (see illustration in fig. 7). This also leads to matricized grid tensors of rank $r^{N/4 + N/8}$. On the other hand, the hybrid mode tree $H$ tiles $\mathcal{I}$ and $\mathcal{I}^c$ entirely into singletons (see illustration in fig. 7), leading (by theorem 1) to grid tensor matricization ranks of $r^{N/2}$. This means that if we were to replicate grid tensors generated by the tree decomposition of $H$ using those of $T$ or $\bar{T}$ (or a summation thereof), we would need to increase the size constant $r$ super-linearly – by a power of $4/3$ (at least).

The above example can be generalized, by considering swapping the dilations of more than two layers at once. In particular, if $T$ is the mode tree corresponding to the baseline dilated convolutional network (dilation $2^{l-1}$ in layer $l$), $\bar{T}$ is the mode tree corresponding to the network obtained by swapping dilations of groups of $k$ layers (dilation $2^{\lceil l/k \rceil \cdot k - 1 - ((l-1) \bmod k)}$ in layer $l$), and the set of mixture nodes includes all nodes of depth $L-k$, a hybrid mode tree $H$ and an index set $\mathcal{I}$ can be found, such that the tree decomposition of $H$ generates grid tensors whose ranks when matricized w.r.t. $\mathcal{I}$ can only be matched by the tree decompositions of $T$ and $\bar{T}$ if their size constant $r$ is increased by a power of $2/(1 + 2^{1-k})$. Since the mixed decomposition of $T$ and $\bar{T}$ (eq. 4) can realize the tree decomposition of $H$ with double the size constant (claim 1), we conclude that it can, with size constant $2r$, generate grid tensors whose matricization ranks (w.r.t. $\mathcal{I}$) require the tree

---

[4] If $c$ is a scalar and $S$ is a set, $c + S$ stands for the set obtained by adding $c$ to each element in $S$.

decompositions of $T$ and $\bar{T}$ to have size constant $r^{2/(1+2^{1-k})}$ – super-linearly larger. Therefore, in this particular setting, prop. 2 holds and the mixed decomposition of $T$ and $\bar{T}$ is indeed expressively efficient w.r.t. their tree decompositions. Taking into account the fact that the mixed decomposition admits maximal matricization ranks almost always when $g(\cdot)$ is the product operator (see app. G), we formalize the result in network terms and reach corollary 1.

# G   MAXIMALITY OF MATRICIZATION RANKS

In the proof of theorem 1 (app. E.2), and in the derivation of corollary 1 (app. F), we made use of the fact that a tree or mixed decomposition (eq. 3 or 4 respectively), with a product operator $g(\cdot)$, admits maximal matricization ranks almost always. That is to say, for any index set $\mathcal{I} \subset [N]$, the ranks of generated grid tensors $\{\mathcal{A}^y\}_y$ when matricized w.r.t. $\mathcal{I}$, attain their maximum possible values (which depend on both the decomposition and $\mathcal{I}$) for all configurations of weights ($\{\mathbf{a}^{\nu,\gamma,\mathrm{I}}, \mathbf{a}^{\nu,\gamma,\mathrm{II}}\}_{\nu,\gamma}$ for the tree decomposition, $\{\mathbf{a}^{\nu,\gamma,\mathrm{I}}, \mathbf{a}^{\nu,\gamma,\mathrm{II}}\}_{\nu,\gamma}$ and $\{\bar{\mathbf{a}}^{\bar{\nu},\gamma,\mathrm{I}}, \bar{\mathbf{a}}^{\bar{\nu},\gamma,\mathrm{II}}\}_{\bar{\nu},\gamma}$ for the mixed decomposition) but a set of Lebesgue measure zero. Hereinafter we justify this assertion.

When equipped with the product operator ($g(a,b) = a \cdot b$), a tree or mixed decomposition generates grid tensors $\{\mathcal{A}^y\}_y$ whose entries are polynomials in the decomposition weights. Therefore, for any index set $\mathcal{I} \subset [N]$, the entries of the matricizations $\{\llbracket \mathcal{A}^y \rrbracket_{\mathcal{I}}\}_y$ are, too, polynomials in the decomposition weights. Claim 2 below implies that for a particular index $y$, the rank of $\llbracket \mathcal{A}^y \rrbracket_{\mathcal{I}}$ is maximal almost always, *i.e.* for all weight settings but a set of measure zero. Since the union of finitely many zero measure sets is itself a zero measure set (see Jones (2001) for example), we conclude that the ranks of $\{\llbracket \mathcal{A}^y \rrbracket_{\mathcal{I}}\}_y$ are jointly maximal almost always, which is what we set out to prove.

**Claim 2.** *Let $D, M_1, M_2 \in \mathbb{N}$, and consider a polynomial function mapping weights $\boldsymbol{\alpha} \in \mathbb{R}^D$ to matrices $A(\boldsymbol{\alpha}) \in \mathbb{R}^{M_1 \times M_2}$ ("polynomial" here means that all entries of $A(\boldsymbol{\alpha})$ are polynomials in $\boldsymbol{\alpha}$). Denote $R = \max_{\boldsymbol{\alpha} \in \mathbb{R}^D} rank(A(\boldsymbol{\alpha}))$, and consider the set $S := \{\boldsymbol{\alpha} \in \mathbb{R}^D : rank(A(\boldsymbol{\alpha})) < R\}$. This set has Lebesgue measure zero.*

*Proof.* We disregard the trivial case where $R = 0$. Let $\boldsymbol{\alpha}_0$ be a point at which $R$ is attained ($rank(A(\boldsymbol{\alpha}_0)) = R$), and assume without loss of generality that the top-left $R \times R$ minor of $A(\boldsymbol{\alpha}_0)$, *i.e.* the determinant of $A(\boldsymbol{\alpha}_0)_{1:R,1:R}$, is non-zero. The function $p : \mathbb{R}^D \to \mathbb{R}$ defined by $p(\boldsymbol{\alpha}) = \det(A(\boldsymbol{\alpha})_{1:R,1:R})$ is a polynomial, which by construction does not vanish everywhere ($p(\boldsymbol{\alpha}_0) \neq 0$). The zero set of a polynomial is either the entire space, or a set of Lebesgue measure zero (see Caron and Traynor (2005) for proof). Therefore, the zero set of $p(\cdot)$ has Lebesgue measure zero. Now, for every $\boldsymbol{\alpha} \in S$:

$$rank(A(\boldsymbol{\alpha})) < R \implies rank(A(\boldsymbol{\alpha})_{1:R,1:R}) < R \implies p(\boldsymbol{\alpha}) := \det(A(\boldsymbol{\alpha})_{1:R,1:R}) = 0$$

$S$ is thus contained in the zero set of $p(\cdot)$, and therefore too, has Lebesgue measure zero.  □

