# OpenReview forum: "Boosting Dilated Convolutional Networks with Mixed Tensor Decompositions"
_ICLR.cc/2018/Conference — Accept (Oral)_

### Official Review · AnonReviewer3 · 2017-11-25
**see detailed review below.**

**Rating:** 7
**Confidence:** 4

**Review:**

This paper theoretically validates that interconnecting networks with different dilations can lead to expressive efficiency, which indicates an interesting phenomenon that connectivity is able to enhance the expressiveness of deep networks. A key technical tool is a mixed tensor decomposition, which is shown to have representational advantage over the individual hierarchical decompositions it comprises.

Pros:

Existing work have focused on understanding the depth of the networks and established that deep networks are expressively efficient with respect to shallow ones. On the other hand, this paper focused on the architectural feature of connectivity. The problem is fundamentally important and its theoretical development is solid. The conclusion is useful for developing new tools for deep network design.

Cons:

In order to show that the mixed dilated convolutional network is expressively efficient w.r.t. the corresponding individual dilated convolutional network, the authors prove it in two steps: Proposition 1 and Proposition 2. However, in the proof of Proposition 2 (see Theorem 1), the authors only focus on a particular case of convolutional arithmetic circuits, i.e., $g(a,b)= a*b$. In the experiments, see line 4 of page 9, the authors instead used ReLU activation $g(a, b)= max{a+b, 0}$. Can authors provide some justifications of such different choices of activation functions? It would be great if authors can discuss how to generate the activation function in Theorem 1 to more general cases.

---

> ### Author Response · Authors · 2017-12-04
> **Authors' response to review 3**
>
> We thank reviewer for the feedback!
>
> As stated in footnote 9, one may adapt our treatment of Proposition 2 to a different activation (choice of $g(a,b)$) by deriving a result analogous to theorem 1, i.e. by establishing upper and lower bounds on matricization ranks brought forth by a tree decomposition with the respective operator $g(a,b)$.  Such bounds were derived in [1] for the choice $g(a,b)=max{a+b,0}$ corresponding to ReLU activation.  However, since [1] only treats specific mode trees T and index sets I, its bounds cannot be readily used in place of theorem 1.
>
> In terms of our experiments, we present results for the setting $g(a,b)=max{a+b,0}$ (ReLU) merely due to its popularity in practice.  The exact same trends occur under the choice $g(a.b)=a*b$ (convolutional arithmetic circuits).  We have added a footnote to the paper indicating this.
>
> [1] Cohen and Shashua.  Convolutional Rectifier Networks as Generalized Tensor Decompositions.  ICML 2016.

---

### Official Review · AnonReviewer1 · 2017-11-28
**Very interesting and thorough paper.**

**Rating:** 9
**Confidence:** 4

**Review:**

To date the theoretical advantage of deep learning has focused on the concept of "expressive efficiency" where one network must grow much larger to replicate functions that another "more efficient" network can produce. This has focused so far on depth (i.e. shallow networks have to grow much larger than deeper networks to express the same set of networks)

The authors explore another dimension here, namely that of "connectivity". They study dilated convolutional networks and show that intertwining two dilated convolutional networks A and B at various stages (formalized via mixed tensor decompositions) it is more expressively efficient than not intertwining.

The authors' experiments support their theory showing that their mixed strategy leads to gains over a vanilla dilated convolutional net.

I found the paper very well written despite its level of mathematical depth (the authors provide many helpful pictures) and strongly recommend accepting this paper.

---

> ### Author Response · Authors · 2017-12-04
> **Authors' response to review 1**
>
> We thank reviewer for the support!

---

### Official Review · AnonReviewer4 · 2017-12-12
**A step forward for understanding contemporary deep nets**

**Rating:** 8
**Confidence:** 3

**Review:**

(Emergency review—I have no special knowledge of the subfield, and I was told a cursory review was OK, but the paper was fascinating and I ended up reading fairly carefully)

This paper does many things. It adds to a series of publications that analyze deep network architectures as parameterized decompositions of intractably large tensors (themselves the result of discretizing the entire input-output space of the network), this time focusing on the WaveNet architecture for autoregressive sequence modeling. It shows (first theoretically, then empirically) that the WaveNet's structural assumption of a single (perfect) binary tree is holding it back, and that WaveNet-like architectures with more complex mixed tree structures perform better.
Throughout the subject is treated with a high level of mathematical rigor, while relegating proofs and detailed walkthrough explanations to lengthy appendices which I did not have time to review.

Some things I noticed:
- The notation used is mostly consistent, except for some variation between dots (e.g., in Eq. 2) and bra-kets (in Fig. 1) for inner product. While I think I'm in the minority here, I'd personally be comfortable with going a little bit further with index notation and avoiding the cohabitation of tensor and vector notation styles by using indices even for dot products; that said, either kind of vector notation (dots or brakets) is certainly acceptable too.
- There are a couple more nomenclature things that might trip up those of us in the deep learning crowd—we're used to referring to "axes" or "dimensions" of a tensor, but the tensor-analysis world apparently says "modes" (and this is called out once in a parenthetical). As "dimension" means something different to tensor folks (what DLers usually call the "size" of an axis), perhaps standardizing on the shared term "axes" would be worthwhile? Not sure if there's a distinction in the tensor world between the words "axis" and "mode."
- The baseline WaveNet is only somewhat well described as "convolutional;" the underlying network unit is not always a "size-2 convolution" (except for certain values of g) and the "size-1 convolutions" that make it up are simply linear transformations. While the WaveNet derives from convolutional sequence architectures (and the choices of g explored in the original paper derive from the CNN literature) it has at least as much in common with recursive/tree-structured network architectures like TreeLSTMs and RNTNs. In fact, the WaveNet is a special case of a recursive neural network with a particular composition function *and a fixed (perfect) binary tree structure.* As this last condition is relaxed in the present paper, making the space of networks under analysis more similar to the traditional space of recursive NNs, it might be worth mentioning this "alternative history" of the WaveNet.
- The choice of mixture nodes in Fig. 3 is a little unfortunate, because it includes all possible mixture nodes and doesn't make it as clear as the text does that a subset of these nodes can be chosen in the general case.
- While I couldn't follow some of Section 5, I'm a little confused that Theorem 1 appears at first glance to apply only to a non-generalized decomposition (a specific choice of g).
- Caffe would not have been my first choice for such a complex, hierarchically structure architecture; I imagine it forced the authors to write a significant amount of custom code.

---

> ### Author Response · Authors · 2017-12-20
> **Authors' response to review 4**
>
> We thank reviewer for the support!  Thank you also for the useful feedback, which will be taken into account in the final version of the manuscript.
>
> Some comments follow:
> - When treating tensors, we currently employ the notations and terms customary in the tensor analysis community (cf. [1]).  Following reviewer's suggestions, we will add to the preliminaries in appendix A a table translating between different terminologies.
> - As reviewer points out, the tensor decomposition framework we use to analyze WaveNet applies more generally to recurrent architectures.  This is a direction analyzed in a follow-up work, which highlights the connections referenced by reviewer.
> - We are currently migrating our code from Caffe to TensorFlow - the latter indeed admits much simpler implementation.
>
> [1] Wolfgang Hackbusch.  Tensor Spaces and Numerical Tensor Calculus.  Springer textbook.

---

### Decision · Program_Chairs · 2018-01-29
**ICLR 2018 Conference Acceptance Decision**

**Decision:**

Accept (Oral)

**Comment:**

This paper proposes improvements to WaveNet by showing that increasing connectivity provides superior models to increasing network size. The reviewers found both the mathematical treatment of the topic and the experiments to be of higher quality that most papers they reviewed, and were unanimous in recommending it for acceptance in the conference. I see no reason not to give it my strongest recommendation as well.